# Toxoplasma effector TgROP1 establishes membrane contact sites with the endoplasmic reticulum during infection

Chahat Mehra[1], Jesús Alvarado Valverde[2,9], Ana Margarida Nogueira Matias[3,4,9], Francesca Torelli[3,4], Tânia Catarina Medeiros[1], Julian Straub [1], James D. Asaki[5], Peter J. Bradley [5,6], Katja Luck [2], Steffen Lawo [7], Moritz Treeck [3,4] & Lena Pernas [1,5,6,8] ✉

Membrane contact sites (MCS) are essential for organelle communication in eukaryotic cells. Pathogens also establish MCS with host organelles, but the mechanisms underlying these interactions and their role in infection remain poorly understood. Here, using a fluorescence sensor and CRISPR-based loss-of-function screening, together with imaging and proteomics, we identify the parasite effector mediating MCS between host endoplasmic reticulum (ER) and the vacuole containing the intracellular parasite *Toxoplasma gondii*. TgROP1 acts as a tether and mimics a canonical FFAT motif to bind the host ER proteins VAPA and VAPB. The loss of VAPA/B abolished host ER–*Toxoplasma* MCS and decreased pathogen growth. These findings indicate that targeting of host MCS tethers is a strategy exploited by pathogens during infection, which could inform future treatment design.

The long-standing view of organelles as autonomous entities has changed dramatically over the past decades. It is now clear that organelles directly communicate and coordinate functions at membrane contact sites (MCS), regions of close membrane apposition tethered by proteins[1]. MCS mediate the bidirectional transport of signalling molecules, coordinate biosynthetic processes and regulate the spatial distribution of organelles[1]. For example, the outer mitochondrial membrane (OMM) import receptor TOM70 interacts with the endoplasmic reticulum (ER) protein IP3R3 for Ca²⁺ transfer from the ER to mitochondria, while ORP1L on endosomes binds ER VAMP-associated protein A/B (VAPA/B) to regulate late endosomal positioning in response to cellular cholesterol levels[2,3]. The physiological importance of MCS is evidenced by human diseases such as Parkinson's and amyotrophic lateral sclerosis that are linked to altered MCS function or mutations in MCS proteins[4,5].

Soon after the discovery of organelle–organelle MCS, similar interactions were described between host organelles and diverse eukaryotic and prokaryotic pathogens[6–10]. Although initially attributed to steric constraints imposed by large pathogen vacuoles, recent findings show that host–pathogen MCS are mediated by protein tethers and may serve as molecular battlegrounds in host–pathogen interactions. For example, effector proteins of the bacterial pathogen *Chlamydia* interact with host VAPs and ceramide transfer protein (CERT) to form MCS between host ER and *Chlamydia* inclusions[11–13]. The loss of CERT or VAPs reduces inclusion size and infectious progeny production, raising the possibility that *Chlamydia* exploits ER MCS to acquire host lipids[13,14]. In the case of the human parasite *Toxoplasma gondii*, the effector mitochondrial association factor 1 (TgMAF1) binds host translocase of the outer mitochondrial membrane protein 70 (TOM70) to tether the parasite vacuole to host mitochondria[15–17]. These host mitochondria–parasite vacuole

[1]Metabolism of Infection Group, Max Planck Institute for Biology of Ageing, Cologne, Germany. [2]Institute of Molecular Biology (IMB) gGmbH, Mainz, Germany. [3]Cell Biology of Host-Pathogen Interaction Laboratory, Gulbenkian Institute for Molecular Medicine, Lisbon, Portugal. [4]The Francis Crick Institute, London, UK. [5]Department of Microbiology, Immunology & Molecular Genetics, University of California, Los Angeles, CA, USA. [6]Molecular Biology Institute, University of California Los Angeles, Los Angeles, CA, USA. [7]Max Planck Institute for Biology of Ageing, Cologne, Germany. [8]Howard Hughes Medical Institute, University of California Los Angeles, Los Angeles, CA, USA. [9]These authors contributed equally: Jesús Alvarado Valverde, Ana Margarida Nogueira Matias. ✉e-mail: lfpernas@mednet.ucla.edu

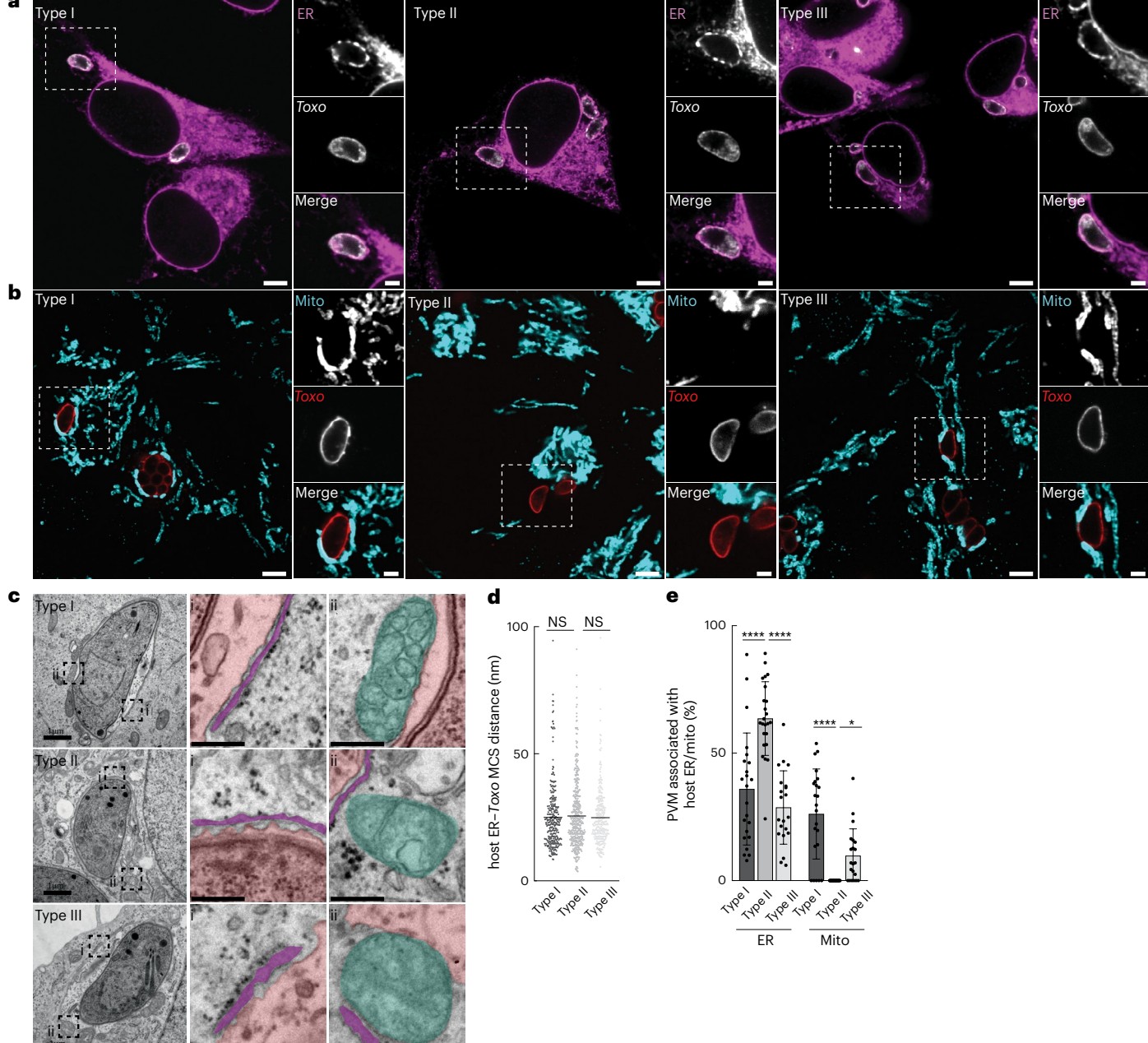

**Fig. 1 | All canonical strains of *Toxoplasma* form MCS with host ER.**
**a**,**b**, Representative immunofluorescence images of (**a**) host endoplasmic reticulum (ER) and (**b**) host mitochondria association with *Toxoplasma* strains type I (RH), type II (ME49) and type III (VEG) in infected ES-2 cells. *Toxo* (surface antigen 1; TgSAG1); ER (calnexin); mito (TOM70). Scale bars, 5 µm (main subpanel) and 2 µm (inset). **c**, Representative electron micrograph images of ES-2 cells infected with indicated *Toxoplasma* strains. MCS between the *Toxoplasma* parasite vacuole membrane (PVM) and (i) host ER and (ii) host mito. Scale bars, 1 µm (main subpanel) and 250 nm (inset). Pink, parasite vacuole; purple, ER; turquoise, mito. **d**, Quantification of the MCS distance between host ER and *Toxoplasma* PVM from images as in **c**. NS, not significant. **e**, Percentage of *Toxoplasma* PVM associated with host ER and mitochondria from infected cells as in **c**. ****$P < 0.0001$; *$P = 0.0209$ by means of one-way ANOVA Tukey's multiple comparison test. Data are mean ± s.d. of $n = 1$ biological replicate (type I, 23; type II, 24; type III, 21 *Toxoplasma* vacuoles).

MCS induce the shedding of large structures positive for OMM, which mediate the depletion of OMM proteins such as mitofusin 1 and mitofusin 2 that restrict fatty acid availability for parasite replication[16,18]. Thus, *Toxoplasma* mitochondria MCS may represent a parasite strategy to counter mitochondrial nutrient competition[19,20].

*Toxoplasma* is the only eukaryotic pathogen reported to form MCS with the host ER[8,9]. However, the function of host ER–*Toxoplasma* MCS remains unknown. This gap in knowledge reflects a broader limitation in our understanding of host–pathogen MCS, which stems from the

lack of identified protein tethers. Traditional biochemical and proteomic approaches used to define tethers are labour intensive, low throughput and have proven insufficient to reliably identify mediators of host–pathogen MCS, underscoring the need for new strategies[21–23].

We adapted a split-green fluorescent protein (GFP) sensor of organelle–organelle MCS to study the MCS that form between *Toxoplasma* and host organelles. By coupling our sensor to fluorescence-activated cell sorting (FACS)-based loss-of-function clustered regularly interspaced short palindromic repeats (CRISPR) screening, we identified

the parasite effector TgROP1 (*Toxoplasma* effector rhoptry protein 1) and the host ER proteins VAPA/B as the protein tethers underlying host ER–*Toxoplasma* MCS. Using live-cell imaging, electron microscopy and proteomics, we show that TgROP1 is required for host ER–*Toxoplasma* MCS and binds VAPA/B at the MSP (major sperm protein) domain known to mediate ER contact sites with other organelles. Last, we show that host VAPs are required for optimal parasite growth. Our results establish TgROP1 as the long-sought tether for host ER–parasite MCS and reveal that pathogens converge on host MCS tethers as a common strategy. Our work paves the way for future studies exploring the factors underlying host–pathogen MCS and their significance during infection.

## All major *Toxoplasma* strains form MCS with host ER

The three predominant lineages of *Toxoplasma*, types I, II and III, differ markedly in their interactions with host cells[24]. For example, MCS between host mitochondria and *Toxoplasma* occurs in a strain-specific manner, as has been reported for the bacterial pathogens *Chlamydia* and *Legionella*[15,25,26]. Previous work exploited these strain-specific differences to identify the parasite mediator of host mitochondria–*Toxoplasma* MCS, termed host mitochondrial association (HMA)[15]. We therefore asked whether the major *Toxoplasma* lineages also differentially associated with host ER. To test this, we examined the interaction between host ER and the *Toxoplasma* parasite vacuole in human ovarian cancer (ES-2) cells infected with representative type I, type II and type III *Toxoplasma* parasites. However, we found that the ER membrane protein calnexin was similarly enriched around all three *Toxoplasma* lineages at 3 h post infection (h.p.i.) (Fig. 1a). As expected, HMA was only observed with types I and III parasites that tether host mitochondria (Fig. 1b)[15].

Organelle–organelle MCS are defined as regions where two membranes are apposed within 10–30 nm (ref. 1). Therefore, to determine whether host ER form MCS with *Toxoplasma*, we performed electron microscopy analysis of ES-2 cells infected with the three canonical strains (Supplementary Fig. 1). In cells infected with type I parasites, the average distance between the host ER and the parasite vacuole membrane (PVM) was 25 nm, consistent with previous reports (Fig. 1c,d)[9]. Type II and type III parasites showed similar distances of 25.5 nm and 24.9 nm, respectively, between host ER and the *Toxoplasma* PVM (Fig. 1d). Ribosomes were excluded from the host ER–*Toxoplasma* interface, as observed at mitochondria–ER MCS (Fig. 1c)[27]. Thus, all major *Toxoplasma* lineages establish MCS with the host ER.

Because types I and III parasites form contact sites with host mitochondria whereas type II parasites do not, we asked whether HMA influenced host ER–*Toxoplasma* interactions[15]. To address this, we compared the extent of host ER–*Toxoplasma* contact sites between type II parasites and types I and III parasites. Whereas type II parasite vacuoles showed extensive contact with host ER—~63% of the parasite vacuole perimeter, in comparison to ~36% and ~29% for types I and III parasites—they lacked detectable contact with host mitochondria, which associated with ~26% and ~10% of type I and type III parasite vacuole perimeters, respectively (Fig. 1c,e). This result suggested that HMA limited the formation of host ER–*Toxoplasma* MCS. To more directly test this possibility, we examined host ER–*Toxoplasma* MCS during infection with WT type I and type I:Δ*maf1* parasites that are deficient for HMA[15]. Indeed, type I:Δ*maf1* parasite vacuoles showed an ~30% increase in host ER MCS relative to WT parasites (Supplementary Fig. 2). Thus, host mitochondria–*Toxoplasma* MCS limit the extent of MCS formed between *Toxoplasma* and host ER.

## A reporter of host–pathogen MCS coupled to loss-of-function CRISPR–Cas9 screening

Our finding that all major strains of *Toxoplasma* form MCS with the ER precluded the use of genetic crosses to map the responsible loci. To overcome this limitation, we sought to develop a sensor of host organelle–*Toxoplasma* MCS that would be amenable to unbiased and high-throughput analyses. To this end, we turned to a split-GFP-based reporter previously established for monitoring interorganelle MCS[28]. In this system, two non-fluorescent moieties of GFP, GFP$^{1-10}$ and GFP$^{β11}$, are targeted to the membranes of distinct organelles. Following the formation of MCS, the GFP is reconstituted and fluoresces[28].

To adapt this reporter for the detection of host–pathogen MCS, we first focused on host mitochondria–*Toxoplasma* MCS, as both host and parasite tethers are known[15–17]. We generated ES-2 cells expressing GFP$^{1-10}$ targeted to the OMM (OMM$^{GFP1-10}$) via the transmembrane domain (TMD) of translocase of the outer membrane 20 (TOM20) (Fig. 2a and Extended Data Fig. 1a)[28]. In parallel, we engineered type I:mCherry-expressing parasites (*Toxo*$^{mCherry}$) to express GFP$^{β11}$ fused to the PVM-targeting N-terminus of TgMAF1 (PVM$^{β11}$) (Fig. 2a and Extended Data Fig. 1b). To account for the ~12 nm distance between host mitochondria and the *Toxoplasma* parasite vacuole, we inserted a 32-amino-acid spacer between the TgMAF1 transmembrane and GFP$^{β11}$ (Fig. 2a)[9,29]. Immunofluorescence analysis of WT and OMM$^{GFP1-10}$-expressing ES-2 cells (OMM$^{GFP1-10}$ ES-2) infected with either *Toxo*$^{mCherry}$ or PVM$^{β11}$-expressing *Toxo*$^{mCherry}$ parasites (*Toxo*$^{PVMβ11}$) revealed GFP exclusively at the host mitochondria–PVM interface when both GFP moieties were present (Fig. 2b). In line with this result, live-cell imaging of primary human foreskin fibroblasts (HFFs) expressing OMM$^{GFP1-10}$ and infected with *Toxo*$^{PVMβ11}$ parasites showed GFP at the mitochondria–*Toxoplasma* interface following the formation of MCS (Fig. 2c and Supplementary Video 1). To test whether our sensor was compatible with high-throughput approaches, we analysed OMM$^{GFP1-10}$ ES-2s that were uninfected or infected with *Toxo*$^{mCherry}$ or *Toxo*$^{PVMβ11}$ parasites

**Fig. 2 | A sensor to identify host organelle–*Toxoplasma* MCS. a**, Schematic of the PVM$^{β11}$ and OMM$^{GFP1-10}$ constructs generated for the host mitochondria–*Toxoplasma* split-GFP system. SP, signal peptide; TM, transmembrane domain. **b**, Representative immunofluorescence images of WT ES-2 cells infected with parasites expressing PVM$^{β11}$ (*Toxo*$^{PVMβ11}$) (left), OMM$^{GFP1-10}$-expressing ES-2 cells (OMM$^{GFP1-10}$ ES-2) infected with mCherry-expressing *Toxoplasma* (*Toxo*$^{mCherry}$) (centre) and OMM$^{GFP1-10}$ ES-2 cells infected with *Toxo*$^{PVMβ11}$ (right). Data are representative of *n* = 2 biological replicates. PVM$^{β11}$ (HA); OMM$^{GFP1-10}$ (myc). Scale bars, 5 μm (main subpanel) and 2 μm (inset). **c**, Live cell images of a HFF expressing OMM$^{GFP1-10}$, infected with *Toxo*$^{PVMβ11}$ parasite and labelled with MitoTracker Deep Red at indicated time points (Supplementary Video 1); white arrowheads indicate GFP at mitochondria–*Toxoplasma* interface. Scale bar, 5 μm. **d**, WT AAVS1 and *TOM70 KO* HeLa cells expressing OMM$^{GFP1-10}$ were infected with *Toxo*$^{PVMβ11}$ and analysed at 8 and 24 h.p.i. by means of flow cytometry for GFP expression. Data are mean ± s.d. of *n* = 5 biological replicates. ****P < 0.0001 for WT AAVS1 versus *TOM70 KO* HeLa cells by two-way ANOVA Sidak's multiple comparison test. **e**, Left: schematic of CRISPR screen to identify parasite mediators of host mitochondria–*Toxoplasma* MCS. Type I *Toxoplasma* expressing PVM$^{β11}$ were transfected with an sgRNA library targeting *Toxoplasma* effector proteins. The resulting pool of KO parasites were used to infect OMM$^{GFP1-10}$ ES-2 cells. At approximately 24 h.p.i., infected cells were sorted based on mCherry and GFP. Right: exemplary infected cells from GFP$^{neg}$ and GFP$^{hi}$ populations obtained during test sorts. Scale bar, 10 μm. **f**, Volcano plot showing the log$_2$FC (*x* axis) and RRA score (*y* axis) of genes from GFP$^{hi}$ versus GFP$^{neg}$ MAGeCK analysis. Genes with log$_2$FC < −0.05 (≥2 guides per gene) are coloured as indicated. **g**, Representative immunofluorescence images of OMM$^{GFP1-10}$ ES-2 cells infected with *Toxo*$^{PVMβ11}$ or Δ*maf1* parasites engineered to express PVM$^{β11}$ (Δ*maf1 Toxo*$^{PVMβ11}$). Data are representative of *n* = 2 biological replicates. PVM$^{β11}$ (HA); OMM$^{GFP1-10}$ (myc). Scale bars, 5 μm (main subpanel) and 2 μm (inset). **h**, Cells infected as in **g** were collected at 8 h.p.i. and 24 h.p.i. and analysed by flow cytometry for GFP expression. WT (*Toxo*$^{PVMβ11}$); Δ*maf1* (Δ*maf1 Toxo*$^{PVMβ11}$). Data are mean ± s.d. of *n* = 3 biological replicates. ****P < 0.0001 for *Toxo*$^{PVMβ11}$ versus Δ*maf1 Toxo*$^{PVMβ11}$ by two-way ANOVA Sidak's multiple comparison test. Panels **a** and **e** were created with BioRender.com.

by flow cytometry. Consistent with our immunofluorescence data, GFP was detected in OMM[GFP1–10] ES-2s infected with *Toxo*[PVMβ11], but not *Toxo*[mCherry] parasites (Extended Data Fig. 2b,c). Thus, our host–pathogen MCS sensor reports on mitochondria–*Toxoplasma* MCS in microscopy and high-throughput approaches.

A known caveat of split-GFP systems is the irreversible nature of GFP reconstitution, which can in some contexts force MCS[1]. To test whether our system artificially induced *Toxoplasma*–host mitochondria contact sites, we compared infections in WT HeLa cells and HeLa cells deficient for TOM70 (TOM70 KO), the host factor required for HMA[16,17]. Electron microscopy showed that *Toxo*[mCherry] and *Toxo*[PVMβ11] parasites established host mitochondria–PVM MCS to a similar extent in WT HeLa cells (Extended Data Fig. 3a,b). No MCS were detected in infected TOM70 KO cells (Extended Data Fig. 3a,b). In line with these results, we found that GFP was detected at the host mitochondria–PVM interface only in WT but not *TOM70 KO* cells expressing OMM[GFP1–10] as assessed by confocal microscopy and flow cytometry (Fig. 2d and Extended Data Fig. 3c). Thus, our host–pathogen MCS sensor

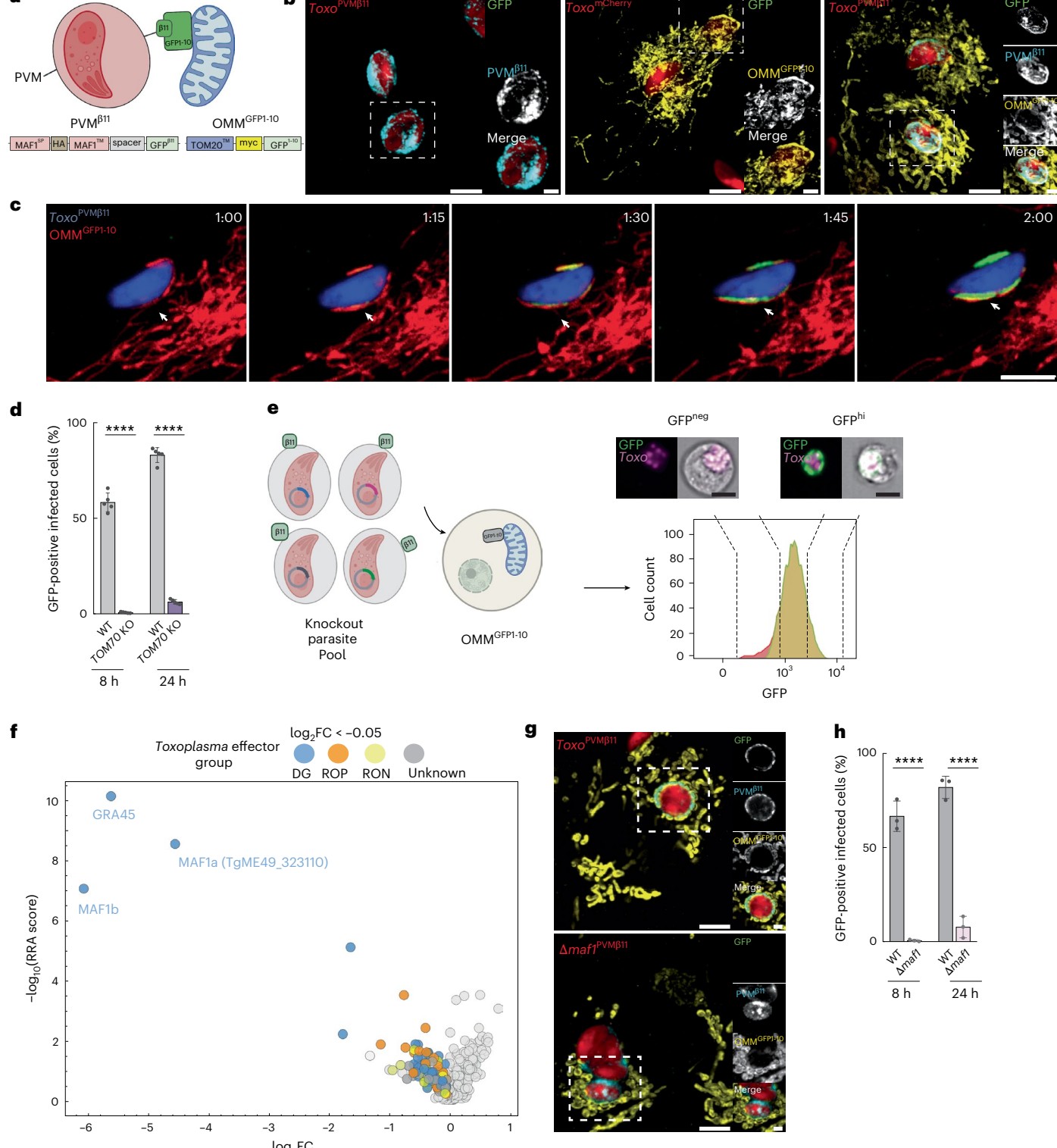

recapitulates the biology of *Toxoplasma*-mitochondria MCS without artificially inducing them.

We next asked whether our sensor could be coupled to unbiased loss-of-function approaches to identify the tethers required for host ER–*Toxoplasma* MCS. To this end, we turned to a previously established CRISPR library containing single guide RNAs targeting 325 predicted *Toxoplasma* effector proteins (Supplementary Table 1)[30,31]. *Toxoplasma* effectors are derived from two secretory organelles: rhoptries and dense granules, which contain rhoptry neck proteins (RONs), rhoptry bulb proteins (ROPs) and dense granule proteins (GRAs)[32]. We chose this library because a subset of ROPs and GRAs localize to the PVM, making them ideal candidates for MCS tethers, as in the case of the dense granule effector TgMAF1[15,32]. To establish proof-of-principle CRISPR–associated protein 9 (Cas9) screening with our sensor, we transfected the *Toxoplasma* effector sgRNA library into parasites expressing the PVM$^{β11}$ construct (Fig. 2e). The resulting pool of mCherry-expressing knockout (KO) parasites was then used to infect OMM$^{GFP1–10}$ ES-2s at a multiplicity of infection (MOI) of 0.5 to decrease the probability of multiply infected cells and enable the assessment of the role of individual *Toxoplasma* effectors in HMA (Fig. 2e).

At 24 h.p.i. when contact site formation was maximal, mCherry-positive, infected cells were FACS-sorted into GFP-negative (GFP$^{neg}$) and GFP-high (GFP$^{hi}$) populations (Fig. 2e). We reasoned that sgRNAs enriched in the GFP$^{neg}$ population but depleted in the GFP$^{hi}$ population would correspond to genes required for host mitochondria–*Toxoplasma* MCS (Fig. 2e). To identify these genes, we extracted *Toxoplasma* genomic DNA (gDNA) and amplified *Toxoplasma* sgRNAs from the GFP$^{neg}$ and GFP$^{hi}$ populations for next-generation sequencing (Fig. 2e). Using the model-based analysis of genome-wide CRISPR–Cas9 KO (MAGeCK) method, we quantified the median log$_2$ fold change (log$_2$FC) in sgRNA abundance between the GFP$^{hi}$ and GFP$^{neg}$ populations and ranked genes using robust rank aggregation (RRA) (Fig. 2f)[33]. This analysis identified TgMAF1, dense granule protein 45 (TgGRA45) and TgME49_323110 (TgMAF1a) as the top candidate promoters of HMA (Fig. 2f and Supplementary Table 2). These results were expected as TgMAF1 binds TOM70 to mediate HMA[15–17]. Indeed, OMM$^{GFP1–10}$ ES-2s infected with Δ*maf1* parasites engineered to express PVM$^{β11}$ were mostly GFP negative at 8 h.p.i. and 24 h.p.i. (Fig. 2g,h). TgGRA45 is a chaperone-like protein required for the PVM localization of GRAs such as TgMAF1[34]. Last, the TgMAF1 locus comprises multiple *TgMAF1* gene copies that belong to two distinct TgMAF1 paralogues: TgMAF1a (including TgME49_323110) and TgMAF1b[35]. Although TgMAF1b but not TgMAF1a tethers mitochondria, the targeting of TgME49_323110 likely disrupted the entire *MAF1* locus[35]. Thus, our host–pathogen MCS reporter is compatible with unbiased and high-throughput loss-of-function approaches.

## CRISPR-based discovery of host ER–*Toxoplasma* MCS

Having validated our split-GFP sensor using host mitochondria–*Toxoplasma* MCS, we next adapted it to study host ER–*Toxoplasma* MCS, for which the molecular tethers remain unknown[8,9]. To do so, we target GFP$^{1–10}$ to the host ER membrane by fusing it to the TMD

of the ER phosphatase suppressor of actin mutations 1 (ERM$^{GFP1-10}$) (Extended Data Fig. 4a,b)[28]. GFP was only detected at the host ER–*Toxoplasma* interface (Extended Data Fig. 4c). Moreover, at 24 h.p.i., greater than 40% of ERM$^{GFP1-10}$-expressing ES-2s infected with *Toxo*$^{PVMβ11}$ were GFP positive, unlike *Toxo*$^{mCherry}$-infected cells which were mostly GFP negative (Extended Data Fig. 4d). Thus, our split-GFP sensor can be adapted to analyse diverse host organelle–pathogen MCS.

To identify the *Toxoplasma* factor(s) that mediates host ER–*Toxoplasma* MCS, we applied the same experimental pipeline used for the host mitochondria–*Toxoplasma* MCS screen (Fig. 3a). ERM$^{GFP1-}$ES-2s were infected with PVM$^{β11}$-expressing parasites transfected with the effector sgRNA library (Fig. 3a). At 24 h.p.i., mCherry-positive infected cells were FACS-sorted into GFP$^{hi}$ and GFP$^{neg}$ populations (Fig. 3a). As before, we reasoned that sgRNAs enriched in GFP$^{neg}$ cells but depleted in the GFP$^{hi}$ cells would target candidate mediators of host ER–*Toxoplasma* MCS, as was the case for TgMAF1 (Fig. 2f). MAGeCK analysis of *Toxoplasma* sgRNA abundances in the sorted GFP$^{hi}$ and GFP$^{neg}$ populations identified TgGRA45 as a top hit (Fig. 3b and Supplementary Table 3)[33]. Because TgGRA45 is required for the PVM insertion of TgMAF1, and thus the correct targeting of PVM$^{β11}$, its enrichment was expected (Extended Data Fig. 5a)[34]. Indeed, Δ*gra45* parasites were completely deficient for host mitochondria–*Toxoplasma* MCS but showed no defect in host ER–PVM MCS (Extended Data Fig. 5b,c) Thus, host ER–*Toxoplasma* MCS form independently of TgGRA45.

## The TgROP1 is required for host ER–*Toxoplasma* MCS

The finding that the loss of TgGRA45 did not impair host ER–*Toxoplasma* MCS allowed us to exclude PVM-localized GRAs as candidate mediators and instead indicated a possible role for rhoptry proteins. As aforementioned, rhoptry effectors can be subdivided into RONs and ROPs. Because RONs are required for host cell attachment and invasion and remain localized at the host plasma membrane, they are unlikely mediators of host ER–*Toxoplasma* MCS[32]. By contrast, ROPs are secreted during invasion, with a subset localizing to the PVM[32]. We therefore focused on ROP effectors with a log$_2$FC < −0.05 (≥2 guides per gene), which yielded 12 candidates (Fig. 3c). Reasoning that an ER-tethering factor should be localized to the PVM, we found that 7 of the 12 ROP candidates encoded predicted TMDs or were known to localize to the PVM (Fig. 3c)[36,37]. Because *Toxoplasma* targets TOM70, a known interorganelle MCS tether, to form MCS with host mitochondria, we also expected that *Toxoplasma* might similarly exploit ER resident proteins that mediate MCS with multiple organelles. To this end, we focused on host VAPA and VAPB, which bind two phenylalanines in an acidic track (FFAT) motif-containing proteins through their MSP domain[27,38].

We next screened the seven ROP candidates for the presence of the VAP-interacting FFAT motif. As a starting point, we analysed our candidates for the presence of the canonical FFAT motif (EFFDAxE) from the Eukaryotic Linear Motif (ELM) database[39]. This yielded one match in the ROP6 protein sequence (Table 1). Given that the spacing of flanking acidic residues can vary across motif instances, we broadened the consensus FFAT motif to ExFxDAxE, thereby allowing for variation

**Fig. 3 | TgROP1 tethers host ER to the parasite vacuole. a**, Left: schematic of CRISPR screen. Type I *Toxoplasma* expressing GFP$^{β11}$ were transfected with an sgRNA library targeting *Toxoplasma* effectors. The resulting pool of KO parasites was used to infect ERM$^{GFP1-10}$ ES-2 cells. ERM, endoplasmic reticulum membrane. At 24 h.p.i., cells were sorted based on mCherry and GFP; right: exemplary infected cells and GFP$^{neg}$ and GFP$^{hi}$ populations obtained during test sorts. Scale bar, 10 μm. **b**, Volcano plot showing the log$_2$FC (*x* axis) and RRA score (*y* axis) of genes from GFP$^{hi}$ versus GFP$^{neg}$ MAGeCK analysis. Genes with log$_2$FC < −0.05 (≥2 guides per gene) are coloured as indicated. **c**, AlphaFold multimer model of the MSP domain of VAPA with the TgROP1 ExFxDAxE motif. **d**, Representative immunofluorescence images of ES-2 cells infected with WT, Δ*rop1:ROP1-HA* and

Δ*rop1* parasites at 3 h.p.i. Data are representative of *n* = 1 biological replicate. PVM (MAF1). Scale bar, 2 μm. **e**, Representative electron micrograph images of ES-2 cells infected with WT (*Toxo*$^{mcherry}$), Δ*rop1:ROP1-HA* and Δ*rop1* parasites at 3 h.p.i. MCS between the *Toxoplasma* PVM and (i) host ER and (ii) host mito. Scale bars, 1 μm (main subpanels) and 250 nm (insets). Pink, parasite vacuole; purple, ER; turquoise, mito. **f**, Percentage of *Toxoplasma* PVM associated with host ER and mitochondria in images as in **e**. Electron microscopy data are mean ± s.d. from *n* = 2 biological replicates (WT, 64; Δ*rop1*, 68; Δ*rop1:ROP1-HA*, 61 -*Toxoplasma* vacuoles). ****$P$ < 0.0001 by means of one-way ANOVA Tukey's multiple comparison test.

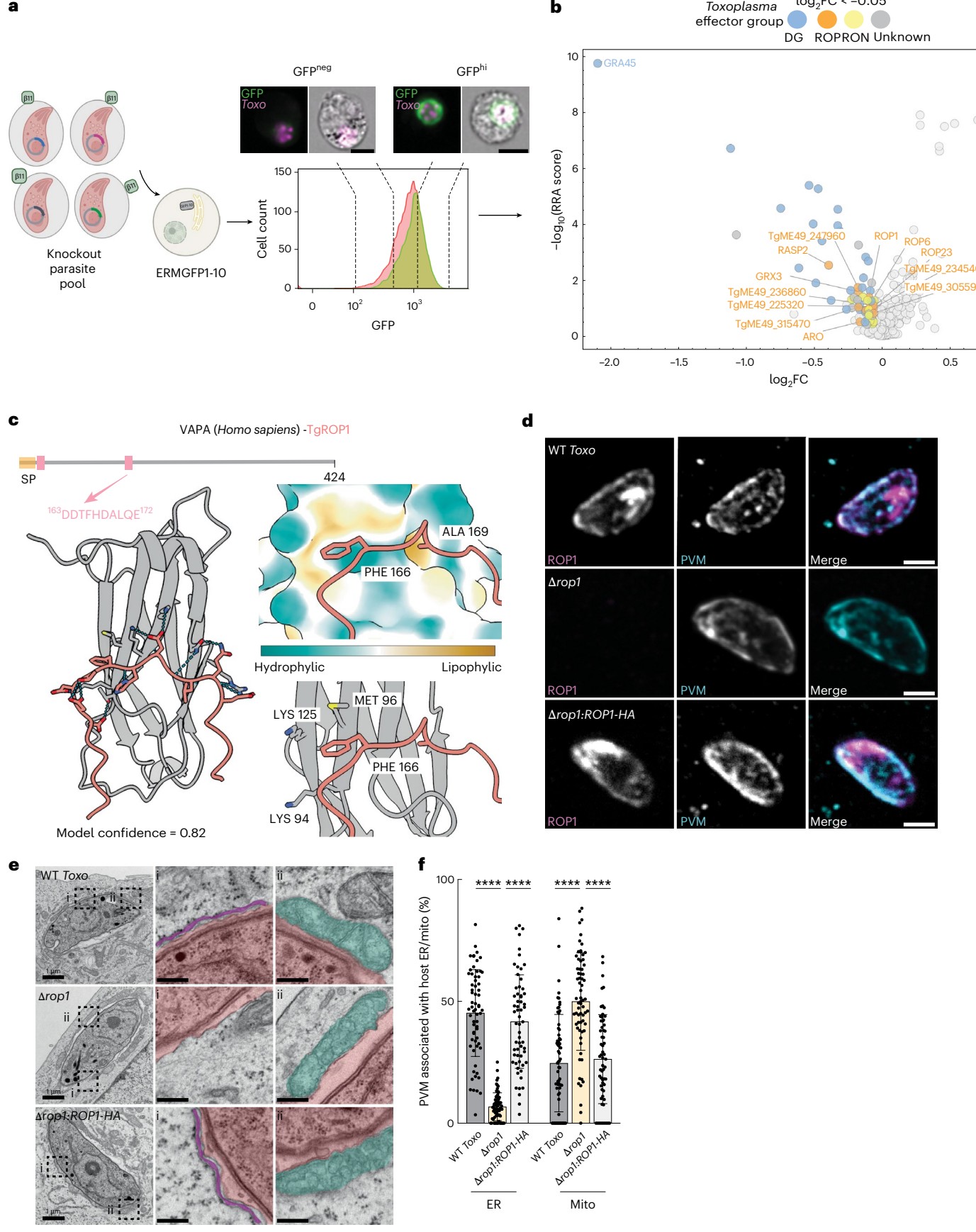

**Table 1 | Top *Toxoplasma* ROP candidate mediators of ER–*Toxoplasma* MCS**

| *Toxoplasma* ID | Hyperlopit description | TMD/PVM | FFAT motif(s) | VAPA interaction |
|---|---|---|---|---|
| TGME49_315160 | rhoptry apical surface protein (RASP2) | No | - | - |
| TGME49_247960 | hypothetical protein | Yes | No | - |
| TGME49_247580 | glutaredoxin 3 (GRX3) | No | - | - |
| TGME49_315470 | hypothetical protein | Yes | Yes (1) | Low |
| TGME49_261440 | armadillo repeats only protein (ARO) | No | - | - |
| **TGME49_258660** | **rhoptry protein 6 (ROP6)** | **Yes** | **Yes (2)[a]** | **High** |
| **TGME49_309590** | **rhoptry protein 1 (ROP1)** | **Yes** | **Yes (2)** | **High** |
| TGME49_225320 | Hypothetical protein | No | - | - |
| TGME49_239600 | rhoptry protein 23 (ROP23) | No | - | - |
| TGME49_234540 | Hypothetical protein | Yes | No | - |
| TGME49_236860 | Haloacid dehalogenase family hydrolase domain-containing protein | Yes | Yes (1) | Low |
| TGME49_305590 | ABC transporter transmembrane region domain-containing protein | Yes | Yes (1) | Low |

TMD/PVM indicates the presence of a predicted TMD and/or known PVM localization. FFAT motif (EFFDAxE or ExFxDAxE) indicates the presence and number of motifs predicted for each gene. VAPA interaction indicates the confidence of VAPA-interaction prediction (Methods). Bold text indicates rhoptry candidates that fulfil all criteria. [a] EFFDAxE.

in distance from the core motif while following the guidelines of ELM[40]. Using this relaxed motif definition, we identified putative FFAT motifs in five of the seven TMD-containing ROPs, including ROP6 (Fig. 3c, Table 1 and Supplementary Table 4).

To evaluate whether these motifs could interact with VAPA, we used AlphaFold Multimer to model the MSP domain of VAPA together with the motif-containing fragments from each ROP candidate. Consistent with previous reports that using peptides fragments rather than full-length proteins improves AlphaFold specificity, we generated models of high confidence for select candidates[41]. Model confidence was assessed by overall model interface predicted template modeling (ipTM) and predicted template modeling score (pTM) confidence (0.8ipTM + 0.2pTM), motif fragment pLDDT (predicted local distance difference test) and predicted aligned error, in line with previously established thresholds[41]. Candidates with model confidence $\geq 0.7$ and average motif pLDDT $\geq 70$ were classified as high confidence[41]. Candidates below these thresholds were considered low confidence (Fig. 3c and Supplementary Table 4). Two ROPs were high confidence: TgROP6 and TgROP1.

TgROP6 contained both a canonical EFFDAxE motif (residues 88–89) and a modified ExFxDAxE motif (residues 323–332), with AlphaFold model scores of 0.82 and 0.74 and corresponding motif pLDDT values of 79.4 and 86.89, respectively (Extended Data Fig. 6a and Supplementary Table 4). To address the role of TgROP6 in host ER–*Toxoplasma* MCS, we compared the association between host ER and the PVM of WT type I *Toxoplasma* parasites or type I parasites engineered to be deficient for TgROP6 (Δ*rop6*) (Extended Data Fig. 6b). However, we observed no significant differences in host ER–*Toxoplasma* MCS between WT parasites and Δ*rop6* parasites (Extended Data Fig. 6c,d). Thus, TgROP6 is dispensable for host ER–*Toxoplasma* MCS.

We next focused on TgROP1, which contained a high-confidence VAP-interacting ExFxDAxE motif at residues 163–172 with a model confidence score of 0.82 and a motif pLDDT value of 88.55 (Supplementary Table 4). Structural modelling predicted that Phe166 and Ala169 of TgROP1 occupied the two hydrophobic pockets characteristic of canonical FFAT–MSP interactions (Fig. 3d). The DDTFHDALQE

motif of TgROP1—which we confirmed localized to the PVM—is conserved across canonical *Toxoplasma* strains that formed host ER–*Toxoplasma* MCS (Figs. 1 and 3e and Extended Data Fig. 7a)[37].

To test whether TgROP1 was required for host ER–*Toxoplasma* MCS, we examined ES-2 cells infected with *Toxoplasma* type I WT and TgROP1 KO parasites (Δ*rop1*) by electron microscopy[37]. Δ*rop1* vacuoles showed an ~80% decrease in ER association and a corresponding increase in HMA relative to WT parasites (Fig. 3f,g). Complementation of Δ*rop1* parasites with Tg*ROP1-HA* (Δ*rop1:ROP1-HA*) restored host ER association to WT levels (Fig. 3f,g). Similar results were obtained with type II WT, Δ*rop1*, and Δ*rop1:ROP1-HA* parasites (Extended Data Fig. 7b,c). Thus, TgROP1 is the major parasite factor required for host ER–*Toxoplasma* MCS.

## Host VAPA/B are required for ER–*Toxoplasma* MCS

To test whether TgROP1 interacted with VAPA/B as predicted by our structural analyses, we immunopurified TgROP1 from cells infected with Δ*rop1:ROP1-HA* parasites. Immunoblot analysis revealed that VAPA and VAPB, but not the ER membrane proteins calnexin nor HMGCR, were enriched in TgROP1-HA immunoprecipitates (Fig. 4a). Furthermore, neither MAF1 nor TOM70 co-immunoprecipitated with TgROP1-HA, indicating that VAPA/B enrichment was not an indirect consequence of nonspecific binding to PVM factors or association with PVM bound mitochondria (Fig. 4a). To conversely identify interacting partners of GFP–VAPA[WT] in an unbiased manner, we next performed proteomics of VAPA-GFP immunoprecipitate from *Toxo[mcherry]*-infected cells. TgROP1 emerged as the most abundant *Toxoplasma* interactor of VAPA (Fig. 4b). Thus, TgROP1 interacts with VAPA/B.

Having established that TgROP1 interacts with VAPA/B, we next asked whether these host factors mediated *Toxoplasma*–ER MCS. As protein tethers are often enriched at MCS, we examined the distribution of VAPA/B during infection[1]. Live-cell imaging of HFFs stably expressing GFP–VAPA[WT] and infected with type I *Toxoplasma* parasites revealed the enrichment of VAPA around the parasite vacuole soon after invasion (Fig. 4c and Supplementary Video 2). Similarly, in VAP double-knockout (DKO) HeLa cells that were reconstituted with GFP–VAPA[WT] or GFP–VAPB[WT], both VAPA and VAPB were found enriched at the parasite vacuoles of types I and II parasites (Fig. 4d,e, Extended Data Fig. 8 and Supplementary Fig. 3). The redistribution of VAPA/B was completely dependent on TgROP1; neither VAPA nor VAPB accumulated at the Δ*rop1* parasite vacuoles (Fig. 4d,e, Extended Data Fig. 8 and Supplementary Fig. 3). Thus, both VAPA and VAPB are sequestered at the *Toxoplasma* vacuole in a TgROP1-dependent manner.

To determine whether VAPA/B are required for host ER–*Toxoplasma* MCS, we compared host ER enrichment around the *Toxoplasma* parasite vacuole in WT and VAP DKO HeLas[42]. Using calnexin as an ER marker, confocal microscopy revealed that ER association with the parasite vacuole was abolished in VAP DKO cells (Supplementary Fig. 4). To assess more precisely the effect of VAP ablation on host ER–*Toxoplasma* MCS, we examined WT and VAP DKO cells infected with type I parasites by electron microscopy. The loss of VAPA/B resulted in a 90% decrease in MCS between host ER and the *Toxoplasma* vacuole (Fig. 4f,g). Conversely, HMA was increased in VAP DKO cells, supporting that host ER–*Toxoplasma* MCS constrain contact sites between host mitochondria and the *Toxoplasma* parasite vacuole (Fig. 4f,g). Thus, VAPA and VAPB are the host factors required for ER–*Toxoplasma* MCS.

## *Toxoplasma* exploits the MSP domain of host VAPs

The loss of VAPA/B led to a significant reduction in parasite growth, raising the possibility that TgROP1 evolved to mimic interactors of VAPA/B to enable *Toxoplasma* exploitation of MCS with host ER (Fig. 4h). To test this, we generated Δ*rop1* parasites expressing a mutant of TgROP1 predicted to be deficient for VAPA/B binding due to a point mutation in

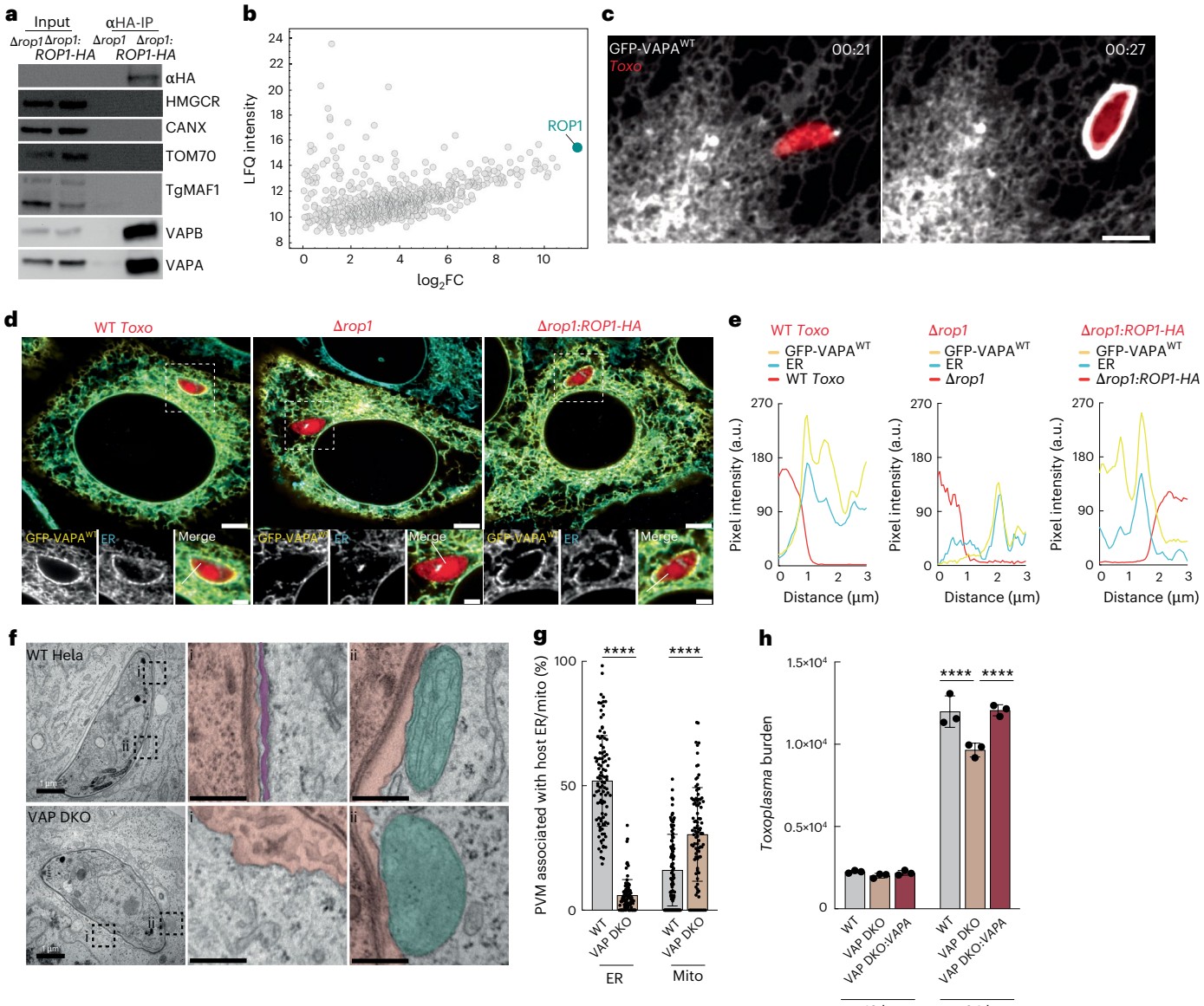

**Fig. 4 | VAPA and VAPB are the host factors that mediate host ER–*Toxoplasma* MCS. a**, Anti-HA immunoprecipitates from WT HeLas infected with Δ*rop1* and Δ*rop1:ROP1-HA* parasites were analysed by means of immunoblotting for VAPA, ~33 kDa; VAPB, ~33 kDa; calnexin, ~90 kDa; HMGCR, ~95 kDa; TOM70, ~72 kDa; TgMAF1, ~55 kDa; HA, ~55 kDa. **b**, Anti-GFP immunoprecipitates from VAP DKO cells expressing GFP–VAPA^WT that were uninfected (uninf) or infected with *Toxo*^mCherry and analysed by mass spectrometry; data represent *Toxoplasma* proteins that had a positive log₂FC. LFQ, label-free quantification. **c**, Representative live-cell confocal microscopy images of HFFs expressing GFP–VAPA^WT and infected with type I *Toxoplasma* at indicated times after infection (Supplementary Video 2). Data are representative of *n* = 1 biological replicate. Scale bar, 5 μm. **d**, Representative immunofluorescence images of VAP DKO HeLa cells expressing GFP–VAPA and infected with the type I WT (*Toxo*^mcherry), Δ*rop1* and Δ*rop1:ROP1-HA* parasites at 3 h.p.i. Data are representative of *n* = 2

biological replicates. ER (calnexin). Scale bars, 5 μm (main subpanel) and 2 μm (inset). **e**, Corresponding pixel intensity plots for white line in the insets in **d**. **f**, Representative electron microscopy images of WT and VAP DKO HeLa cells infected with type I *Toxoplasma* at 3 h.p.i. MCS between the *Toxoplasma* parasite vacuole and (i) host ER and (ii) host mito. Scale bars, 1 μm (main subpanel) and 250 nm (inset). Pink, parasite vacuole; purple, ER; turquoise, mito. **g**, Percentage of *Toxoplasma* PVM associated with host ER and mitochondria in images as in **e**. Electron microscopy data are mean ± s.d. from *n* = 3 biological replicates (WT, 103; VAP DKO, 96 *Toxoplasma* vacuoles). ****P < 0.0001 by means of two-tailed unpaired *t*-test. **h**, WT, VAP DKO and VAP DKO HeLas expressing GFP–VAPA^WT were infected with *Toxo*^mCherry parasites and analysed at 24 h.p.i. by flow cytometry for *Toxoplasma* burden (mCherry median fluorescence intensity). FI, fluorescence intensity. Data are mean ± s.d. from *n* = 3 infected wells of one biological replicate ****P < 0.0001 by means of one-way ANOVA analysis.

---

Phe166 (Δ*rop1:ROP1*^F166A), a residue that AlphaFold-modelling indicated was critical for binding the MSP domain of VAPA/B (Fig. 3d). To examine its effect on ER association, GFP–VAPA^WT expressing VAP DKO cells were infected with either Δ*rop1*, Δ*rop1:ROP1-HA*^WT or Δ*rop1:ROP1*^F166A-HA parasites. The ROP1^F166A mutant localized to the PVM similarly to WT ROP1 (Extended Data Fig. 9). Δ*rop1:ROP1-HA*^F166A vacuoles failed to enrich GFP–VAPA^WT, phenocopying Δ*rop1* parasites (Fig. 5a,b). Thus, the ROP1 FFAT-like motif is necessary for its interaction with VAPA.

To determine whether TgROP1 binds the MSP domain of VAPA/B, we examined ER association in VAP DKO cells expressing either GFP–VAPA^WT or the MSP mutant GFP–VAPA^K94D/M96D, which is deficient for binding FFAT-containing proteins. In GFP–VAPA^K94D/M96D-expressing VAP DKO cells, neither GFP–VAPA^K94D/M96D nor calnexin were enriched at the *Toxoplasma* vacuole (Extended Data Fig. 10a,b). Consistent with this result, electron microscopy analysis revealed that GFP–VAPA^K94D/M96D cells were deficient for host ER–*Toxoplasma* MCS

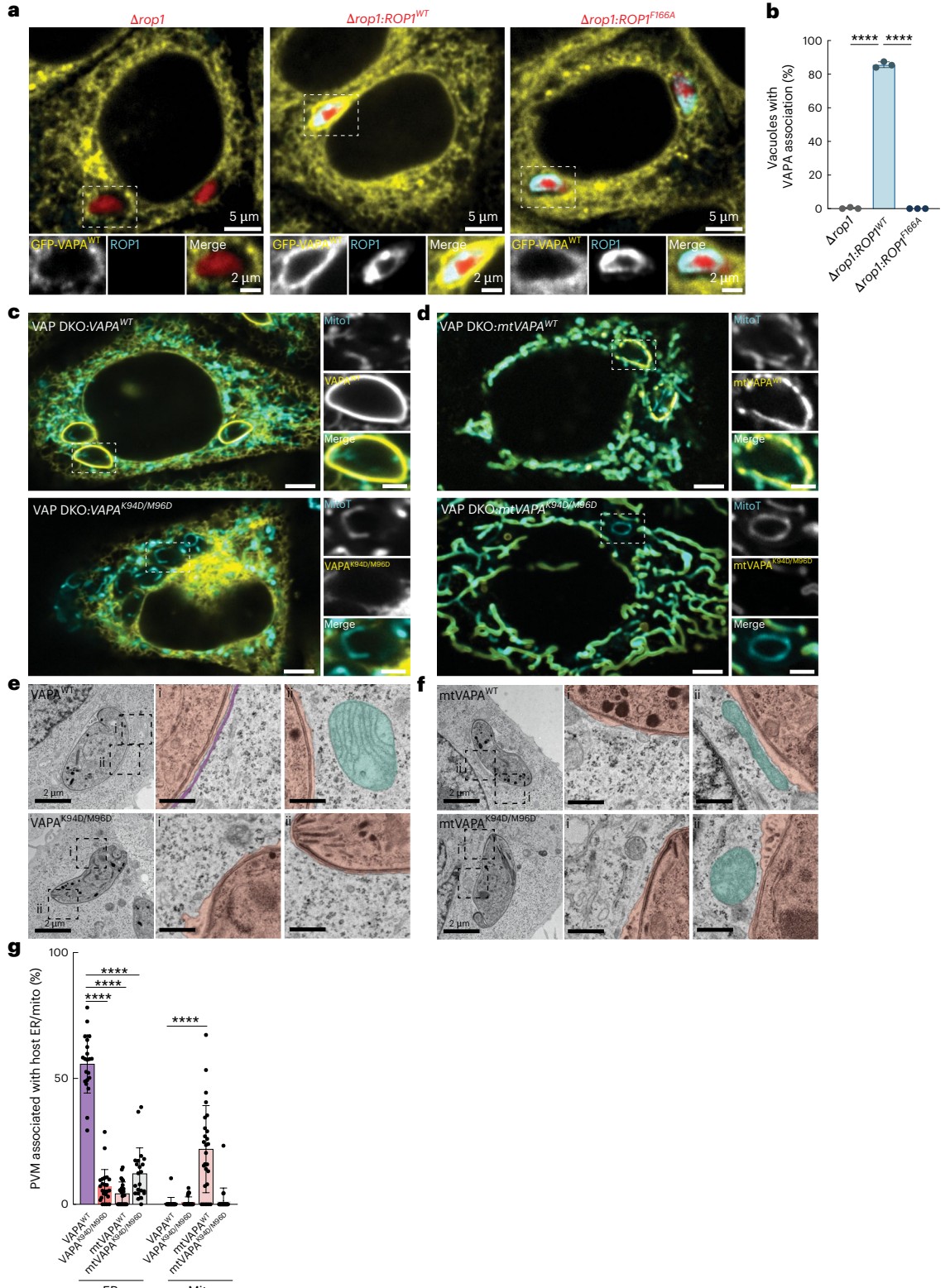

**Fig. 5 | *Toxoplasma* exploits the MSP domain of VAPA to form host ER–*Toxoplasma* MCS. a**, Representative immunofluorescence images of VAP DKO HeLas expressing VAPA-GFP infected with type I Δ*rop1*, Δ*rop1:ROP1*$^{WT}$-*HA* and Δ*rop1:ROP1*$^{F166A}$-*HA* parasites at 3 h.p.i. **b**, Percentage of vacuoles showing VAPA-GFP association from images as in **a**. Data are mean ± s.d. from *n* = 3 biological replicates; ****$P < 0.0001$ by means of one-way ANOVA. **c,d**, Representative live-cell confocal images of VAP DKO expressing GFP–VAPA$^{WT}$ and GFP–VAPA$^{K94D/M96D}$ (**c**) or OMM-targeted mtGFP–VAPA$^{WT}$ and mtGFP–VAPA$^{K94D/M96D}$ (**d**) that were infected with type II (Pru) and labelled

with MitoTracker Deep Red. Scale bars, 5 μm (main subpanel) and 2 μm (inset). **e,f**, MCS between the *Toxoplasma* PVM and (i) host ER and (ii) host mito from cells described as in **c** and **d**. Scale bars, 2 μm (main subpanel) and 500 nm (inset). Pink, parasite vacuole; purple, ER; turquoise, mito. **g**, Percentage of *Toxoplasma* PVM associated with host ER and mitochondria in images as in **e**. Electron microscopy data are mean ± s.d. from *n* = 1 biological replicate (VAPA$^{WT}$, 21; VAPA$^{K94D/M96D}$, 24; mtVAPA$^{WT}$, 26; mtVAPA$^{K94D/M96D}$, 22 *Toxoplasma* vacuoles.) ****$P < 0.0001$ by means of one-way ANOVA Tukey's multiple comparison test.

(Extended Data Fig. 10c,d). Similar results were obtained for the corresponding GFP-VAPB[K87D/M89D] MSP mutant (Supplementary Fig. 5)[38].

To exclude the possibility that the loss of *Toxoplasma*-ER MCS in VAP DKO cells expressing VAPA/B MSP mutants was due to pleiotropic effects on the ER membrane, we required an approach that would enable us to address the role of the MSP domain independently of the ER membrane. We therefore leveraged the fact that type II parasites do not tether host mitochondria, and retargeted GFP-VAPA and GFP-VAPA[K94D/M96D] to the OMM by replacing the ER TMD with that of OMP25 (mtVAPA and mtVAPA[K94D/M96D], respectively). Remarkably, mtVAPA[WT] was sufficient to drive MCS between type II parasite vacuoles and host mitochondria as assessed by confocal microscopy and electron microscopy (Fig. 5c–g). By contrast, mtVAPA[K94D/M96D] failed to establish MCS between host mitochondria and type II parasites, analogous to the loss of ER MCS observed in infected VAP DKO cells expressing the VAP MSP mutants (Fig. 5c–g). Thus, TgROP1 exploits the conserved MSP domain of VAPA/B to form host ER–*Toxoplasma* MCS.

## Discussion

In this Article, we developed a sensor to study the MCS that form between *Toxoplasma* and host ER or mitochondria that can be adapted for the study of host–pathogen MCS between any genetically tractable pathogen and host organelle. Furthermore, our sensor is amenable to high-throughput approaches and genetic screening. Coupling this sensor to loss-of-function CRISPR screening in *Toxoplasma*, we discovered that TgROP1 and VAPA/VAPB are the tethers that mediate *Toxoplasma*–host ER MCS.

These findings raise several questions, beginning with the physiological relevance of host ER–*Toxoplasma* MCS. VAPs mediate lipid transfer at ER–organelle MCS, and *Toxoplasma* relies on host-derived lipids to sustain vacuole and plasma membrane biogenesis. Host ER–*Toxoplasma* MCS may therefore facilitate parasite acquisition of host lipids[27]. In line with this possibility, we and others observed that the loss of VAPs restricted *Toxoplasma* growth in HeLa cells[43]. Alternatively, VAPA and VAPB may promote parasite growth in an MCS-independent manner: the loss of TgROP1, the parasite factor that binds VAPA/B, was required for optimal parasite growth in IFNγ-activated macrophages but not in HFFs[37]. Thus, MCS between *Toxoplasma* and host ER may shield the parasite against immune restriction factors that impair PVM integrity.

Our data show that *Toxoplasma* uses effectors from distinct secretory organelles to establish organelle-specific MCS: TgROP1 from rhoptries tethers host ER, and TgMAF1 from dense granules tethers host mitochondria. Rhoptries are released concomitant with *Toxoplasma* invasion. Meanwhile, dense granules are released following invasion and throughout the intracellular life cycle of the parasite. Consistent with this, host ER is observed most readily early after infection. Thus, host organelle–*Toxoplasma* MCS are temporally regulated and may impinge on each other. Indeed, our results indicate that host ER–*Toxoplasma* MCS limit the extent of host mitochondria–*Toxoplasma* MCS and vice versa. This competition may reflect shifting parasite needs: early access to ER-derived host lipids may support vacuole expansion, while tethering mitochondria later may mediate a *Toxoplasma* defence. In line with this, TgMAF1 drives the shedding of OMM proteins, reducing mitochondrial access to nutrients[16,18].

Are other effectors recruited to host ER-*Toxoplasma* MCS? A recent study identified TgVIP1 as a dense granule protein that contains an FFAT motif and partially colocalizes with VAPA[43]. Although loss of TgVIP1 did not effect host ER-*Toxoplasma* MCS at 6 hpi, it slightly reduced their formation at 24 hpi. Thus, other effectors including TgVIP1 may be recruited to host ER-*Toxoplasma* MCS after their establishment, and contribute to their stability.

Why are VAPA, VAPB and TOM70 targeted by diverse pathogens including *Toxoplasma, Chlamydia* and SARS-CoV-2, respectively?[27,44]. One commonality between VAPA/B and TOM70 is that these proteins are key mediators of host organelle–organelle contact sites. *Chlamydia* effectors interact with host VAPs and the lipid transfer protein CERT to form MCS between host ER and *Chlamydia* inclusions[11–13]. Genetic ablation of CERT or VAP leads to smaller *Chlamydia* inclusions and decreases the production of infectious progeny[13,14]. Conversely, human cytomegalovirus virus infection decreases in ER–mitochondria MCS and thus stimulator of interferon genes (STING)-dependent immune signalling[45]. Thus, by interacting with MCS mediators, pathogens may benefit from the various functions of MCS such as lipid transfer or disrupt the organelle–organelle MCS that enable immune signalling transduction.

Our development of a sensor for host–pathogen MCS led to the identification of TgROP1 and VAPA/B as the only known tethers mediating MCS between host ER and a eukaryotic pathogen. This discovery reveals that pathogens exploit host contact site tethers during infection and paves the way for the systemic dissection of host–pathogen MCS function during infection.

## Methods

### Mammalian cell culture

HeLa adenocarcinoma cells, ES-2 ovary clear cell carcinoma cells and HFF cells were obtained from the American Type Culture Collection (CCL-2, CRL-1978 and SCRC-1041, respectively); VAPA/B DKO cells were a kind gift from Dr. Pietro Di Camelli[42]. All cells were cultured at 37 °C and 5% CO$_2$ in Dulbecco's modified Eagle's GlutaMAX medium and supplemented with 10% heat-inactivated FBS (Gibco A3840402) and 100 U ml$^{-1}$ penicillin–streptomycin (Thermo Fisher Scientific 15070063) (referred to as cDMEM). Cells were routinely tested for *Mycoplasma* infection by polymerase chain reaction (PCR).

### Cloning

For stable expression of plasmids, the triple haemagglutinin tag (3XHA-) enhanced green fluorescent protein (eGFP) and outer mitochondria membrane protein OMP25 targeting sequence pMXs-3XHA-eGFP-OMP25 (Addgene 83356) was always used and modified as discussed. For split-GFP constructs, a complementary DNA containing myc-GFP1-10 gene strand was synthesized by Eurofins Genomics. This was digested with restriction enzymes BamHI and NotI and inserted into pMXs-3XHA-eGFP vector (pMXs-myc-GFP1-10). For creation of OMM-targeted GFP1-10, the TOM20 N-terminal targeting sequence was added to primers 1 and 2, and a PCR was performed with the pMXs-myc-GFP1-10 plasmid[28]. The PCR product was treated with the Kinase, Ligase, and DpnI enzyme mix (NEB M0554S) according to manufacturer's protocol. To create ERM-targeted GFP1-10, the Sac1 ER targeting sequence was amplified from a plasmid provided by Dr. Cali using primers 3 and 4 with the forward primer containing a myc tag and subsequently inserted into pMXs-myc-GFP1-10 with restriction enzymes XhoI and NotI (pMXs_myc_ER)[28]. pMXs_myc_ER was further modified by inserting GFP (amplified from pMXs-3XHA-eGFP-OMP25) or GFP1-10 (amplified from pMXs-TOM20-myc-GFP1-10) with primers 5 and 6 and primers 7 and 8, respectively, using restriction enzymes BamHI and XhoI with HiFi DNA assembly cloning.

Human VAPA and VAPB cDNA was amplified from ES-2 cells with primers 9 and 10 and primers 11 and 12, respectively, and inserted into the pMXs-3XHA-eGFP plasmid backbone with restriction enzymes SacII and NotI to create pMXs-3xHA-GFP-VAPA (GFP-VAPA[WT]) and pMXs-3xHA-GFP-VAPB (GFP-VAPB[WT]). To create VAPA K94D/M96D mutant (GFP-VAPA[K94D/M96D]) and VAPB K87D/M89D mutant (GFP-VAPB[K87D/M89D]), the plasmids pMXs-3xHA-GFP-VAPA and pMXs-3xHA-GFP-VAPB were modified using primers 13 and 14 and primers 15 and 16, respectively. The PCR products were treated with the KLD enzyme. For creation of the mitochondrially targeted VAPA constructs, the plasmids GFP-VAPA[WT] and GFP-VAPA[K94D/M96D] were modified by primers 17 and 18 to remove the VAPA transmembrane and replace it with the transmembrane of OMP25 (obtained from pMXs-3XHA-eGFP-OMP25

(Addgene 83356) plasmid) in one PCR reaction. The PCR product was treated with KLD enzyme.

For creation of *Toxoplasma* split-GFP constructs, the previously described N-terminally tagged MAF1 expression construct was modified to only retain the HA tag and N-terminus of the MAF1 sequence until the MAF1 transmembrane domain[15]. The PCR product was treated with KLD enzyme to remove unmodified plasmid (pMAF1_N-term). A 32-amino-acid spacer $\beta_{11}$ was subsequently inserted with restriction enzymes XhoI and NotI using the primers 19 and 20 (PVM$^{\beta11}$)[29]. PVM$^{\beta11}$ plasmid was further modified with primers 21 and 22 to insert chloramphenicol selection cassette via ClaI and BclI enzymes[16]. To create the ROP1 point mutation (ROP1$^{F166A}$), the F amino acid at position 166 was mutated to an A, the pUPRT-RH ROP1-HA plasmid[37] was modified with primers 23 and 24 and the PCR product was treated with the KLD enzyme. All plasmids were verified by sanger sequencing performed at Eurofins Genomics. All primer sequences are listed in Supplementary Table 6.

### Lentiviral production

For production of lentivirus, human embryonic kidney 293T cells were transfected using the X-tremeGENE 9 DNA Transfection Reagent (Roche) with the following plasmid combination: 1 μg UMVC (Gag-pol) packaging vector, 0.3 μg pCMV-VSVG envelope vector (Addgene 8454) and 1 μg of the relevant plasmid of interest. The next day, the medium of each well was changed, and 2 days after transfection, the virus-containing supernatant was filtered through a 0.45 μm filter and supplemented with polybrene (Sigma TR-1003) to a final concentration of 5 μg ml$^{-1}$. The virus-containing filtrate was added to 50,000 target cells and exchanged for cDMEM the next day. ES-2, HeLa and HFF cells were subsequently selected with 10–18 μg ml$^{-1}$ blasticidin for 3–5 days.

### Parasite culture and generation of parasite strains

All parasite strains were maintained by serial passage on HFF monolayers in cDMEM. *Toxoplasma* strains used in this study include the following: type I (RHΔ*hxgprt*), type II (ME49Δ*hxgprt*:mScarlet)[46] and type III (VEG) strains (deleted for the *hypoxanthine–xanthine–guanine phosphoribosyl transferase* (*HXGPRT*) gene); type I RHΔ*KU80*Δ*hxgprt*; type I Δ*gra45*; type I RHΔ*hxgprt*:mCherry+ (*Toxo*$^{mCherry}$); type I RHΔ*maf1:mCherry* + (Δ*maf1*); and RHΔ*maf1-HA-MAF1:mCherry* + (Δ*maf1*-HA-MAF1)[15,16,34]. For TgROP1 characterization, the following parasites were used: type I *Toxoplasma*−RHΔ*rop1* and RHΔ*rop1:ROP1-HA*; and type II *Toxoplasma*−PruΔ*KU80* Δ*hxgprt* (type II WT), PruΔ*rop1* and PruΔ*rop1:ROP1-HA*[37]. All strains were routinely tested for *Mycoplasma* infection by PCR.

To create transgenic parasites, RHΔ*hxgprt* (for CRISPR screen), RHΔ*hxgprt*:mCherry+ (*Toxo*$^{PVMβ11}$) and RHΔ*maf1:mCherry* + (Δ*maf1 Toxo*$^{PVMβ11}$) were transfected with 50–60 μg of the PVM$^{\beta11}$ plasmid following BglIII-linearization and then selected with 20 μM chloramphenicol (Sigma R4408). Two weeks after selection, the populations were cloned out via serial dilution. Single clones were confirmed with hemagglutinin (HA) staining in immunofluorescence assays.

To create ROP6 KO parasites, a protospacer targeting the coding region of ROP6 was introduced into the pCas9-GFP:sgRNA CRISPR plasmid (generated using primers 25 and 26) via KLD cloning. For ROP6, Pro$^{GRA1}$-mCherry-T2A-HXGPRT-Ter$^{GRA2}$ construct was amplified from a template plasmid using primers 27 and 28 to introduce a 40 base pair homology to the 5′ and 3′ untranslated regions of ROP6[31]. Approximately 5 μg of the PCR product and 30 μg of plasmid were co-transfected into type I *Toxoplasma* strain RHΔ*KU80*Δ*hxgprt* and selected with 25 μg ml$^{-1}$ mycophenolic acid (Sigma 475913) and 50 μg ml$^{-1}$ xanthine (Alfa Aesar A11077) for 1 week before the populations were cloned out and ROP6 KO was confirmed with immunofluorescence and western blot using anti-ROP6 murine monoclonal antibody.

To create the ROP1 mutant parasites (Δrop1:ROP1$^{F166A}$-HA), 2 μg of the pUPRT-RH ROP1 $^{F166A}$-HA plasmid was linearized and transfected together with 2 μg pCas9 plasmid targeting UPRT, into the RHΔROP1 parasites using the AMAXA Nucleofector 4D system. The next day the parasites were treated with 5 μM of 5′-fluo-2′-deoxyuridine and subsequently cloned out. Single clones were verified for integration with PCR and immunofluorescence.

### Live-cell imaging

Cells were seeded on 24-well CELLview glass bottom cell culture plates (Greiner Bio-One) and imaged using an Olympus IXplore SpinSR 50 mm spinning disk confocal microscope. Live-cell imaging was performed with incubation at 37 °C and 5% CO$_2$. Single-plane or *z*-stack images were taken with a ×100/1.35 silicon oil objective, 488, 561 or 640 laser lines and cellSens Software.

### Immunofluorescence assays and antibodies

ES-2 or HeLa cells were plated in a 24-well glass-bottom plate (Greiner Bio-One) and infected with *Toxoplasma* strains for 3 h.p.i., 8 h.p.i. or 24 h.p.i. as indicated in text. Cells were fixed in 4% paraformaldehyde in prewarmed cDMEM for 15 min at 37 °C, permeabilized for 10 min at room temperature with 0.2% triton (unless indicated otherwise), blocked in 3% BSA in 1xPBS for 20–30 min, and incubated in primary antibodies overnight at 4 °C. After being washed 3 times for 5 min with 1× PBS, cells were incubated in secondary antibody for 40 min to 1 h at room temperature. Plates were rinsed 3 times for 5 min in 1× PBS and maintained in 1× PBS until imaging. For primary antibodies, calnexin (GeneTex GTX109669 [C3], C-term or Proteintech 10427-2-AP); TOMM70 (HPA 048020); HA (Roche (3F10), 11867423001 or CST, 3724); c-Myc (CST 5605S, D84C12) or Myc-tag (Proteintech 16286-1-AP); antisera of TgMAF1[15]; GFP (Takara Bio 632380); TgROP1 (Abnova MAB17504); TgROP6 (mouse monoclonal) were used at 1:300–1:1,000 or 1:2,000 overnight. Secondary antibodies used were the following: Alexa Fluor Plus 405, Alexa Fluor Plus 488, Alexa Fluor Plus 594 and Alexa Fluo Plus 647 (Thermo Fisher). Single-plane or *z*-stack images were taken with a ×60/1.35 or ×100/1.35 silicon oil objective and excitation with either 405, 488, 561 or 640 confocal or Olympus super resolution laser lines with an Olympus IXplore SpinSR spinning disk confocal microscope.

### Electron microscopy sample preparation and analysis

ES-2 and HeLa cells (as indicated in text) were grown on small discs of aclar foil (Science Services E50425-10) in either 24-well or 12-well plates and infected with *Toxoplasma* strains for indicated times. Then the discs were fixed for 1 h in 2% glutaraldehyde (Sigma G5882-100ML) with 2.5% sucrose (Roth 4621.1) and 3 mM CaCL$_2$ (Sigma C7902-500G) in 0.1 M HEPES buffer (Sigma C7902-500G) at pH 7.4. Samples were washed three times with 0.1 M HEPES buffer and incubated with 1% osmium tetroxide (Science Services E19190) and 1% potassium hexacyanoferrate (Sigma P8131) for 1 h at 4 °C. After being washed 3 times for 5 min with 0.1 M cacodylate buffer (Applichem A2140,0100), samples were dehydrated at 4 °C using ascending ethanol series (50%, 70%, 90%, 3× 100%) for 7 min each. Infiltration was performed with a mixture of 50% Epon/ethanol for 1 h, 70% Epon/ethanol for 2 h and with pure Epon (Sigma 45359-1EA-F) overnight at 4 °C. Samples were embedded into TAAB capsules (Agar Scientific G3744) and cured for 48 h at 60 °C. Ultrathin sections of 70 nm were cut using an ultramicrotome (Leica Microsystems UC6) and a diamond knife (Diatome) and stained with 1.5% uranyl acetate (Agar Scientific R1260A) for 15 min at 37 °C and 3% Reynolds lead citrate solution made from lead (II) nitrate (Roth HN32.1) and tri-sodium citrate dehydrate (Roth 4088.3) for 4 min. Images were acquired using a JEM-2100 Plus Transmission Electron Microscope (JEOL) operating at 80 kV equipped with a OneView 4 K camera (Gatan).

For quantification of host–ER *Toxoplasma* MCS, images of *Toxoplasma* parasite vacuoles in infected ES-2 or HeLa cells were analysed using ImageJ software v.2.1.0. To measure the percentage of the PVM associated with host organelles, the total length of contacts between

the organelles was added and divided by the perimeter of the PVM (PVM associated with host ER or mitochondria/total PVM perimeter × 100). In Fig. 1, one pack each from 3 h.p.i. and 24 h.p.i. was analysed. In all other figures, all parasite vacuoles only from the indicated times were assessed.

## Flow cytometry analysis

For split GFP assays, monolayers of infected-ES2 or HeLa cells were rinsed with PBS, trypsinized and fixed in 2% paraformaldehyde in 3% FBS in 1× PBS (FACS buffer) for 5 min. After a spin at 1,000 r.p.m. for 5 min, cells were resuspended in FACS buffer, and a minimum of 10,000 events were analysed on a FACSFortessa using BD FACSDiva software v.8.0.1. The data were then analysed in BD FlowJo software v.10.10.0 as outlined in Extended Data Fig. 2. To assess parasite proliferation, monolayers of ES-2 cells infected with $Toxo^{mCherry}$ parasites were left to grow for 24 h.p.i. and collected as previously described[18]. Then 10,000 events were analysed on a FACSFortessa and the mCherry median fluorescence intensity (mFI) using BD FACSDiva software.

## Creation of the CRISPR parasite pool

A pool of single-stranded DNA oligonucleotides encoding the proto-spacer sequences targeting the *T. gondii* secretome was selected from an arrayed library using an Echo 550 Acoustic Liquid Handler (Labcyte) in three independent events and then pooled to minimize loss of guides. The pooled oligonucleotides were integrated in the pCas9–mCherry–HXGPRT:sgRNA CRISPR vector by Gibson cloning after digestion with PacI/NcoI (NEB), resulting in a library of 1,644 sgRNAs targeting 325 genes, with an average of 5 sgRNAs per gene[30,31]. A total of 180 × 10^6 PVM^β11-expressing parasites were transfected in triplicate with 150 μg of KpnI-linearized (NEB) and phenol-chloroform purified library with the P3 Primary Cell 4D-Nucleofector kit (Lonza V4XP-3032) in a Amaxa 4D Nucleofector (Lonza AAF-1003X, program EO-115). Stable integration of the pCas9–mCherry–HXGPRT:sgRNA library was induced upon treatment with 25 μg ml⁻¹ Mycophenolic acid and 50 μg ml⁻¹ xanthine (Sigma-Aldrich) the following day. An average transfection efficiency of 1.2% corresponding to a coverage of 1,000 parasites per sgRNA was estimated from the parasite survival rate at day 7 after transfection in a plaque assay. Three days after transfection, the selected pool of KO parasites was syringe-lysed and added to fresh HFF monolayers with 100 U ml⁻¹ Benzonase (Merck) overnight to remove traces of input DNA. Seven days after transfection, parasites from individual transfections were pulled and stored in liquid nitrogen in 50 × 10^6 parasite aliquots until use.

## CRISPR screen

To perform the screen with technical duplicates, two vials of the split-GFP screen parasites (each considered as a technical duplicate) were thawed onto two T175 flasks of HFF monolayers. The next day, the media of the flasks were changed to 25 μg ml⁻¹ mycophenolic acid and 50 μg ml⁻¹ xanthine (Sigma-Aldrich). Two days following treatment with selection media, the parasites were expanded by passing 2^6 parasites onto one T175 flask of HFF monolayer. Two days later, 2^6 parasites (to ensure a 1,000× representation of guides) were added to 6 T175 flasks of HFF monolayers. The next day, 300 × 10^6 OMM GFP^1–10 ES-2 cells and ERM GFP^1–10 ES-2 cells were plated in 15 cm dishes (8 × 10^6 to 10 × 10^6 cells per dish). The next morning, split GFP parasites from each technical replicate were used to infect 150 × 10^6 cells of each cell type at a low MOI of 0.5. The plates were rinsed after infection, and then the next day approximately 24 h after infection, cells from each technical replicate were trypsinized with accutase (to avoid clumping) and pooled together into 50 ml falcons. The cells were fixed in 2% PFA for 5 min in FACS buffer with 5% accutase then spun down at 300 *g* for 5 min to get rid of fixative. The cells were distributed into FACS tubes for sorting. The host mitochondria–*Toxoplasma* MCS screen cells were sorted using a BD FACSAria III sorter, and the host ER–*Toxoplasma*

MCS screen cells were sorted using a BD FACSFusion sorter. Gates were drawn to first sort for infected cells (mCherry fluorescence) and then all cells negative for GFP expression (GFP^neg), and the top 20% of the GFP-positive (GFP^hi) populations were sorted for both screens at 4 °C using a 100 μm nozzle with sheath pressure set at 20 p.s.i.; 0.9% NaCl was used as sheath fluid. Cell pellets were stored at −80 °C. Images representative of screen populations obtained during test sorts performed before the screen were acquired on an ImageStream^X MkII imaging cytometer, at ×60 magnification. Single, focused cells were recorded based on their area and aspect ratio values in channel 1 (brightfield) and gradient RMS values >50. Image analysis was performed using IDEAS software v.6.3.23.0 (Cytek Biosciences).

Cell pellets were then de-crosslinked in a solution of 10 mM Tris pH 7.5 and 10 mg ml⁻¹ Proteinase K (Sigma-Aldrich 3115887001) at 55 °C for 24 h. Cells were then lysed with buffer AL (QIAamp DNA Blood Mini Kit) for 2 h, and gDNA was isolated as per manufacturer's protocol. Library samples and genome-integrated *Toxoplasma* sgRNA sequences were amplified by PCR (22 cycles with 2.5 μg of gDNA as input in 100 μl reaction volume) using NEBNext Ultra II Q5 Master Mix (New England BioLabs) with a mix of five different forward primers (primers 29–33) to introduce sequence variability and a reverse primer (primer 34). Afterwards, amplicons were pooled, bead-purified and quantified followed by the introduction of Illumina Nextera adaptors and indices by eight cycles in a second round of PCR. Samples were analysed on an Illumina NovaSeq platform by paired end (2 × 100 bp) sequencing with >3 × 10^7 reads per sample.

## CRISPR screen data analysis

To analyse the screen data, following demultiplexing, raw next-generation sequencing libraries were quality-checked using FastQC version 0.11.8 (ref. 47). Upstream sequences and sgRNA length were used to trim reads with cutadapt (version 4.5). MAGeCK (version 0.5.9.5) count was used to quantify the number of reads per sgRNA[33]. Raw sgRNA counts were median normalized, and MAGeCK test was used to rank sgRNAs and genes (sgRNAs with fewer than 50 read counts in treatment or control samples were excluded from the analysis). The log$_2$FC on a gene level was calculated as follows: log$_2$FC = median [log$_2$(sgRNA read counts in 'GFP^neg' gate + 1) − (sgRNA read counts in 'GFP^hi' gate + 1)]. For gene significance, an α-RRA score was calculated by MaGeCK[33]. Double-sided volcano plots of gene-level log$_2$FCs and RRA scores were created using Instant Clue software v.0.12.2 [48].

## Immunoprecipitation and immunoblotting

Three million HeLa cells were infected with *RH Δrop1* and *RH Δrop1:ROP1-HA* parasites at a MOI of 3. At 3 h.p.i., cells were rinsed twice in chilled 1× PBS, scraped down in chilled 1× PBS supplemented with phosphatase inhibitors (Sigma 4906845001), centrifuged at 1,500 *g* for 5 min, resuspended in lysis buffer (50 mM Hepes–KOH at pH 7.4, 40 mM NaCl, 2 mM EDTA, 1% Triton X-100 and protease and phosphatase inhibitors (Thermo Scientific A32961 and Sigma 4906845001)) for 15 min at 4 °C and centrifuged at 10,000 g for 15 min. Cleared lysates were incubated with either 25 μl magnetic anti-HA-beads (Thermo Scientific 88837) or 25 μl magnetic anti-GFP-nanobodies (Chromotek GTD-20) overnight. The beads were washed 3 times with 1× PBS with phosphatase inhibitors. Afterwards, the samples were eluted from the magnetic beads with 2× SDS buffer by incubating them at 40 °C for 10 min. Samples were processed for gel electrophoresis, and following gel transfer, the membranes were blocked with TBS–0.05% Tween 20 (TBS-T) and 5% milk; the primary antibodies were incubated overnight. Following incubation, blots were washed three times in TBS-T for 15 min and then incubated with horseradish peroxidase (HRP)-conjugated anti-rabbit IgG (CST 7074) or anti-mouse IgG (CST 7076) at a 1:4,000 dilution for 1 h, washed 3 times with TBS-T and developed using a chemiluminescence system (Pierce ECL Western Blotting Substrate or Pierce SuperSignal West Atto Ultimate Sensitivity Substrate; ThermoFisher Scientific). The

following primary antibodies were used: HMGCR (Sigma AMAB90619), TOM70 (HPA048020), HA-HRP (Roche 12013819001), VAPA (Proteintech 15275-1-AP), VAPB (Proteintech 14477-1-AP), calnexin (Proteintech 10427-2-AP) and antisera of TgMAF1.

## Proteomics sample preparation

To prepare samples from immunoprecipitation, on-beads digestion was performed to elute the proteins off the beads. Before adding the elution buffer, the beads were washed with detergent-free buffer (50 mM Tris–HCl pH 7.5) four times to remove any detergents used previously. Then 100 µl of the elution buffer (5 ng µl$^{-1}$ trypsin, 50 mM Tris–HCl pH 7.5, 1 mM Tris (2-carboxyethyl) phosphine), 5 mM chloroacetamide) was added to the beads which were then incubated at room temperature by vortexing from time to time or rotating on a rotator. After 30 min, the supernatant was transferred to a 0.5 ml tube and incubated at 37 °C overnight to ensure a complete trypsin digest. The digestion was stopped the next morning by adding formic acid to the final concentration of 1%. The resulted peptides were cleaned with home-made StageTips. Peptides were separated on a 25 cm, 75 µm internal diameter packed emitter column (Coann emitter from MS Wil, Poroshell EC C18 2.7 µm medium from Agilent) using an EASY-nLC 1200 (Thermo Fisher Scientific). The column was maintained at 50 °C. Buffers A and B were 0.1% formic acid in water and 0.1% formic acid in 80% acetonitrile, respectively. Peptides were separated on a gradient from 4% to 30% buffer B for 19 min at 400 nl min$^{-1}$, followed by a higher organic wash. Eluting peptides were analysed on a QExactive HF mass spectrometer (Thermo Fisher Scientific) in DIA mode. Peptide precursor $m/z$ measurements were carried out at 120,000 resolution in the 400 to 800 $m/z$ range followed by 29 DIA scans with an isolation width of 14 Th and a resolution of 15,000. MS1 and DIA MS2 scans were recorded in centroid mode.

## Proteomics liquid chromatography coupled to tandem mass spectrometry analysis

The raw data were analysed with Spectronaut 16.2 (Biognosys) using default parameters against the reference proteome for human, UP000005640, downloaded in September 2018. Methionine oxidation and protein N-terminal acetylation were set as variable modifications; cysteine carbamidomethylation was set as fixed modification. The digestion parameters were set to 'specific' and 'Trypsin/P', with two missed cleavages permitted. Protein groups were filtered for at least two valid values in at least one comparison group, and missing values were imputed from a normal distribution with a down-shift of 1.8 and standard deviation of 0.3. Differential expression analysis was performed using limma, v.3.34.9 in R[49].

## FFAT motif search and AlphaFold multimer predictions

All *Toxoplasma gondii* sequences were retrieved from *Toxoplasma* database[50]. The canonical FFAT motif (EFFDAxE) Regular Expression (REGEX) model was retrieved from the ELM database with the entry name TRG_ER_FFAT_1 (ref. [39]). The FFAT relaxed REGEX was defined as [EDST].{1,2}[FY].[DEST][ALCFS].{1,2}[EDST] based on other FFAT motif sequences at the ELM database. We changed the distance of the acidic residue in position 1 of the core motif, allowing for any residue at position 3 and adding more hydrophobic residues at position 5. We screened our candidates for either the canonical or modified FFAT motif against the ROP proteins of strain I and found 7 matches in 5 rhoptry candidates. We then made 7 AlphaFold multimer predictions for the 5 rhoptry candidates following the fragmentation approach previously published[41]. To generate the models, we obtained the MSP domain of VAPA and extended sequences of the motif matches. The human VAPA sequence was retrieved from UniProt with the accession Q9P0L0-1 (ref. [51]) The VAPA MSP domain was first defined based on the InterPro boundaries and then manually extended on both flanks to include residues with high pLDDT values, based on the AlphaFold database reference model[52]. The motif matches were extended on both

flanks by 5 residues. We used a local installation of AlphaFold Multimer version 2.3.2 for all domain–motif pairs using the following parameters to produce five models per pair[52]:

```
--model_preset=multimer
--db_preset=full_dbs
--max_template_date=2020-05-14
--num_multimer_predictions_per_model=1
--use_gpu_relax=True
--data_dir = /mnt/storage/alphafold/v232
--bfd_database_path = /mnt/storage/alphafold/v232/bfd/bfd_
    metaclust_clu_complete_id30_c90_final_seq.sorted_opt
--mgnify_database_path = /mnt/storage/alphafold/v232/mgnify/
    mgy_clusters_2022_05.fa
--obsolete_pdbs_path = /mnt/storage/alphafold/v232/pdb_mmcif/
    obsolete.dat
--pdb_seqres_database_path = /mnt/storage/alphafold/v232/
    pdb_seqres/pdb_seqres.txt
--template_mmcif_dir = /mnt/storage/alphafold/v232/pdb_mmcif/
    mmcif_files
--uniprot_database_path = /mnt/storage/alphafold/v232/uniprot/
    uniprot.fasta
--uniref90_database_path = /mnt/storage/alphafold/v232/
    uniref90/uniref90.fasta
--uniref30_database_path = /mnt/storage/alphafold/v232/
    uniref30/UniRef30_2021_03
--use_precomputed_msas=True
```

## Model scoring

Using the ranking_debug json file, the confidence of the highest scored model per pair was extracted. The model confidence is a weighted metric calculated from the pTM and ipTM as follows: confidence = 0.8ipTM + 0.2pTM. Using a model confidence score threshold, values above 0.7 were considered as high confidence and the ones below as low confidence[41]. The average pLDDT value of the core motifs (excluding the 5 residues flank expansions) was further calculated. Taking together both the model confidence and the motif average pLDDT, we ranked the models as follows: models with high model scores above 0.7 and motif average pLDDT value above 70 were considered as high confidence; models with values below any of these thresholds were considered as low confidence[41].

## Line scan analyses

Line-scan analysis of relative fluorescence intensity was performed by measuring pixel intensity across an indicated line using Fiji software.

## Statistical analyses

All statistical analyses were performed using one-way analysis of variance (ANOVA), two-way ANOVA or an unpaired *t*-test in GraphPad Prism 8 software and are indicated accordingly.

## Reporting summary

Further information on research design is available in the Nature Portfolio Reporting Summary linked to this article.

## Data availability

All data are available in the main text or the Supplementary Information, or provided as raw data. Source data are provided with this paper.

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

## Acknowledgements

We thank ToxoDb/EuPathDB team for developing and maintaining this critical community resource, the Max Planck Institute for Biology of Ageing (MPI-AGE) Proteomics facility; the MPI-AGE FACS and Imaging Core for flow cytometry and microscopy support—in particular K.-F. Donahue, L. Schumacher and M. Germer; and the Cluster of Excellence for Aging Research at the University of Cologne (CECAD) electron microscopy facility, in particular, K. Seidel and F. Wolf. We thank all members of the Pernas laboratory for helpful discussions and, in particular, J. F. Stortz, S. Reato and M. T. Sutterlin for technical support. We thank P. De Camilli for sharing VAP DKO cells, T. Calì for split-GFP constructs information, J. P. J. Saeij for Δ*gra45* parasites and M. Blume for the type III parasites. We are thankful to T. Gibson and the curators of the ELM resource. This work was supported by Cologne Graduate School of Ageing (CGA) research (C.M.); European Research Council ERC-StG-2019 852457 (L.P.); Deutsche Forschungsgemeinschaft (DFG) SFB 1218 Project ID 269925409 (L.P.); Packard Fellowships for Science and Engineering (L.P.), Burroughs Wellcome Fund Pathogenesis of Infectious Disease (BWF-PATH) (L.P.), the Howard Hughes Medical Institute (L.P.); the Wellcome Trust (223192/Z/21/Z) (M.T.); the Francis Crick Institute, which receives its core funding from Cancer Research UK (CC2132 and CR2023/030/2132); the UK Medical Research Council (CC2132 and CR2023/030/2132) (M.T.); the Wellcome Trust (CC2132 and CR2023/030/2132 (M.T.); Fundação para a Ciência e Tecnologia (FCT) 2023.06167.CEECIND and UI/BD/154200/2022 (M.T. and A.M.N.M.); National Institute of Health (NIH) RO1 AI123360 (P.J.B.); TO 1349/1-1, DFG (F.T.); and NIH National Institute of General Medical Sciences (NIGMS) Initiative for Maximizing Student Development (IMSD) GM55052 (J.D.A.). Funding for transmission election microscope instrumentation was provided by the DFG under the grant number INST 216/793-1 FUGG.

## Author contributions

Conceptualization: C.M. and L.P. Methodology: C.M., F.T., J.A.V., A.M.N.M., T.C.M., J.S., S.L., M.T. and L.P. Investigation: all authors. Resources: all authors. Funding acquisition: C.M. and L.P. Project administration: L.P. Writing, original draft: C.M. and L.P. Review and editing: all authors. Supervision: L.P.

## Competing interests

The authors declare no competing interests.

## Additional information

**Extended data** is available for this paper at https://doi.org/10.1038/s41564-025-02193-3.

**Correspondence and requests for materials** should be addressed to Lena Pernas.

**Extended Data Fig. 1 | Characterization of the host mitochondria-*Toxoplasma* MCS sensor. a**, Representative immunofluorescence (IF) images of OMM$^{GFP1-10}$-expressing ES-2 cells (OMM$^{GFP1-10}$ ES-2s). OMM$^{GFP1-10}$ (myc); OMM (TOM20). Scale bar: 5 μm. OMM: outer mitochondrial membrane. **b**, Representative IF images of OMM$^{GFP1-10}$ ES-2s infected with parasites expressing PVM$^{β11}$ (*Toxo*$^{PVMβ11}$). PVM$^{β11}$ (HA); PVM (MAF1). PVM: parasite vacuole membrane; MCS: membrane contact sites. Scale bar, including inset, 5 μm. Both are representative of n = 1 biological replicate.

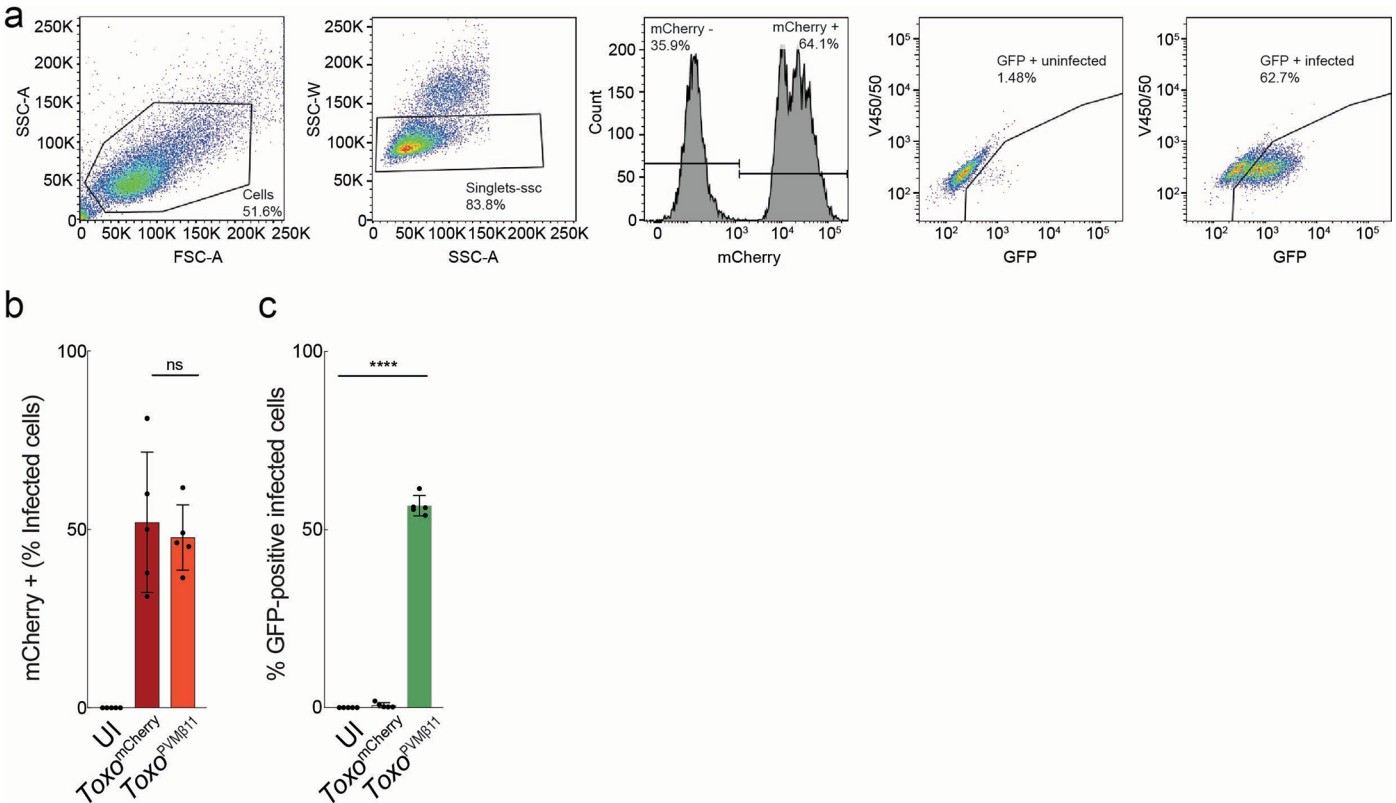

**Extended Data Fig. 2 | Flow cytometry-based analysis host organelle-*Toxoplasma* MCS. a**, Cells were initially gated using forward scatter (FSC) versus side scatter (SSC) to define the population of interest, followed by SSC-width (W) and SSC- area (A) to gate for single cells. The resulting population was then analyzed based on mCherry intensity to distinguish between uninfected (mCherry-; UI) and infected (mCherry + ) cells. The mCherry-positive population was assessed for GFP expression levels. OMM[GFP1–10]-expressing ES-2 cells were either UI, infected with parasites expressing mCherry (*Toxo*[mCherry]), or infected with parasites expressing PVM[β11] (*Toxo*[PVMβ11]) and analyzed by flow cytometry for (**b**) infection (mCherry) and (**c**) GFP expression. PVM: parasite vacuole membrane; OMM: outer mitochondrial membrane. Data are mean ± SD of n = 5 biological replicates. ****p < 0.0001 by means of one-way ANOVA analysis Tukey's multiple comparison test.

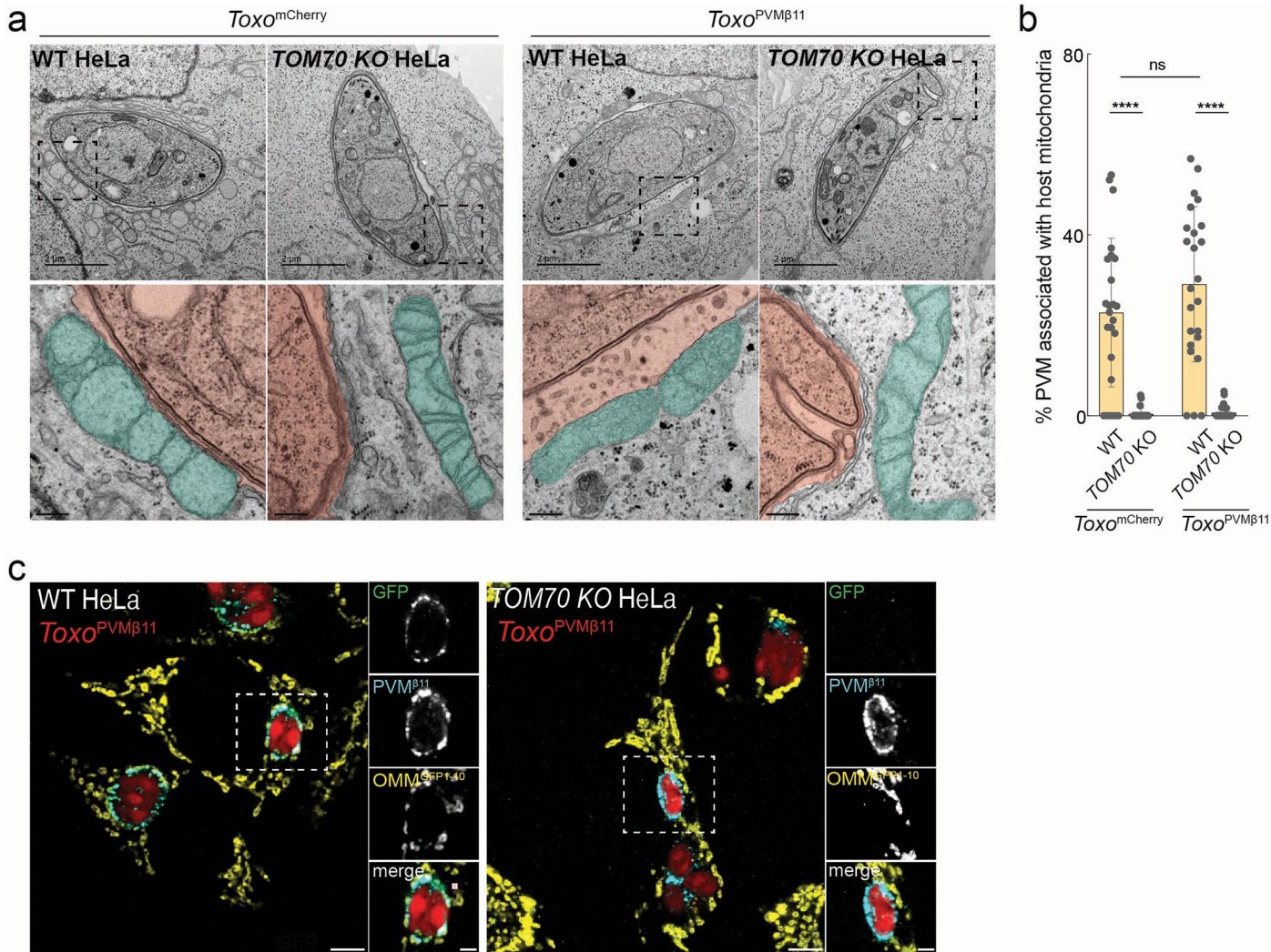

**Extended Data Fig. 3 | Host mitochondria-*Toxoplasma* MCS require TOM70.**
**a**, Representative electron microscopy images of WT OMM[GFP1-10] and *TOM70* KO OMM[GFP1-10] HeLa cells infected with Type I mCherry-*Toxoplasma* (*Toxo*[mCherry]) or parasites expressing PVMβ11 (*Toxo*[PVMβ11]) at 8 hpi. Scale bars: 2 μm; inset, 250 nm. **b**, Membrane contact sites between the *Toxoplasma* parasite vacuole membrane (PVM) and host mito. Pink, parasite vacuole; turquoise, mito. Percentage of *Toxoplasma* PVM associated with host mitochondria in images as in (a) from

n > 20 vacuoles from 1 biological replicate (L-R: 24; 25; 24; 23 *Toxoplasma* vacuoles). ****p < 0.0001 for WT OMM[GFP1-10] versus TOM70 KO OMM[GFP1-10] HeLa cells by two-way ANOVA Sidak's multiple comparison test.
**c**, Immunofluorescence images of WT and *TOM70 KO* HeLas expressing OMM[GFP1-10] that were infected with *Toxo*[PVMβ11]. Data is representative of n = 2 biological replicates. PVM[β11] (HA); OMM[GFP1-10] (myc). PVM: parasite vacuole membrane; OMM: outer mitochondrial membrane. Scale bar: 5 μm; inset, 2 μm.

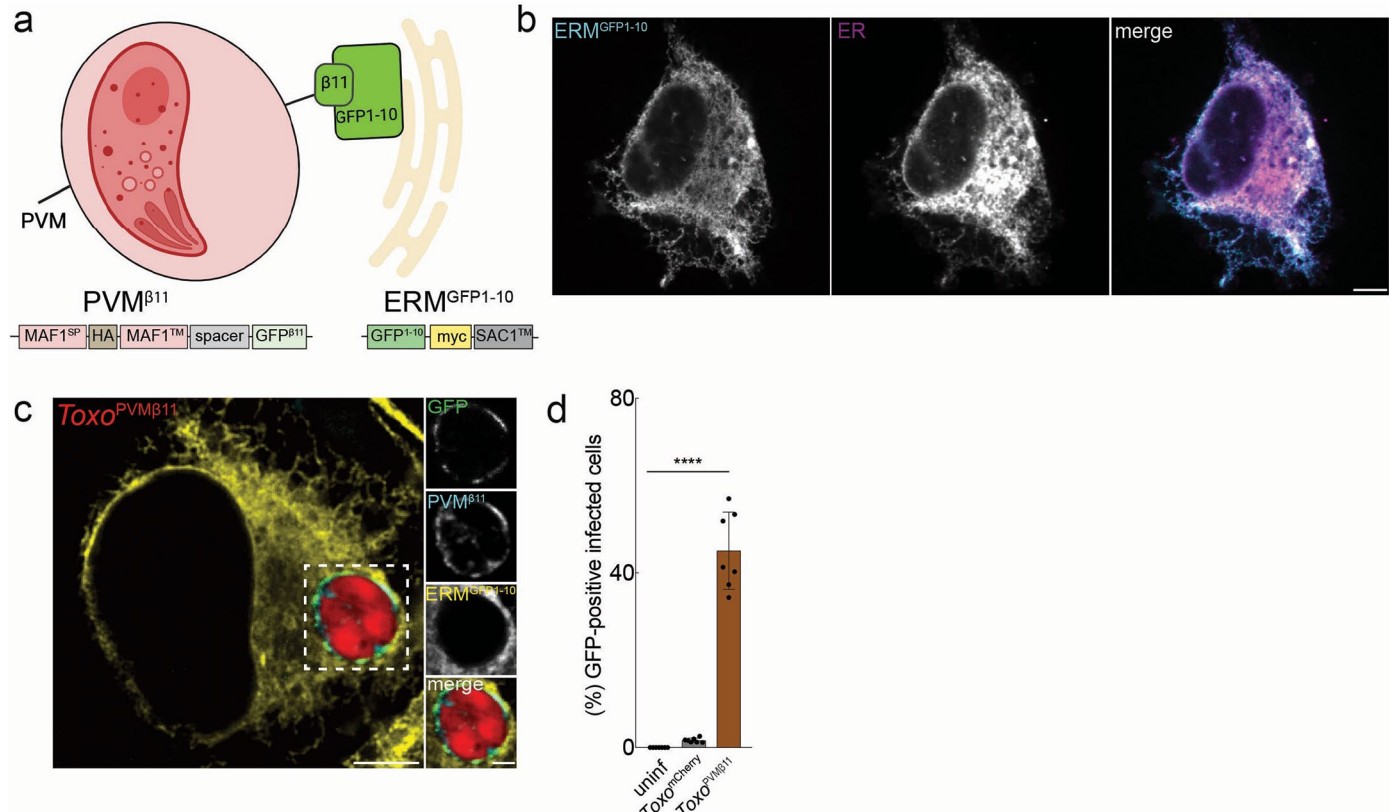

**Extended Data Fig. 4 | Characterization of the host ER-*Toxoplasma* MCS sensor. a**, Schematic of the PVM^β11 and ERM^GFP1-10 constructs generated for the host ER-*Toxoplasma* split-GFP system. PVM, parasite vacuole membrane; SP, signal peptide; TM, transmembrane domain; ERM, ER membrane; SAC1, Phosphoinositide phosphatase. **b**, Representative immunofluorescence (IF) images of ERM^GFP1-10-expressing ES-2 cells (ERM^GFP1-10 ES-2s). ERM^GFP1-10 (GFP); ER (calnexin). Scale bar: 5 mm. **c**, Representative images of ERM^GFP1-10 ES-2s infected with parasites expressing PVM^β11 (*Toxo*^PVMβ11) at 24 h post infection (hpi). Both (**b**) and (**c**) are representative of n = 2 biological replicates. PVM^β11 (HA); ERM^GFP1-10 (GFP). Scale bars: 5 μm; inset, 2 μm. **d**, ERM^GFP1-10 ES-2 cells were uninfected (UI), infected with parasites expressing mCherry (*Toxo*^mCherry) or *Toxo*^PVMβ11 and analyzed by flow cytometry for GFP expression at 24 hpi. FACS data are mean ± SD of n = 4 biological replicates. ****p < 0.0001 for by means of one-way ANOVA using Tukey's multiple comparison test. Panel **a** created with BioRender.

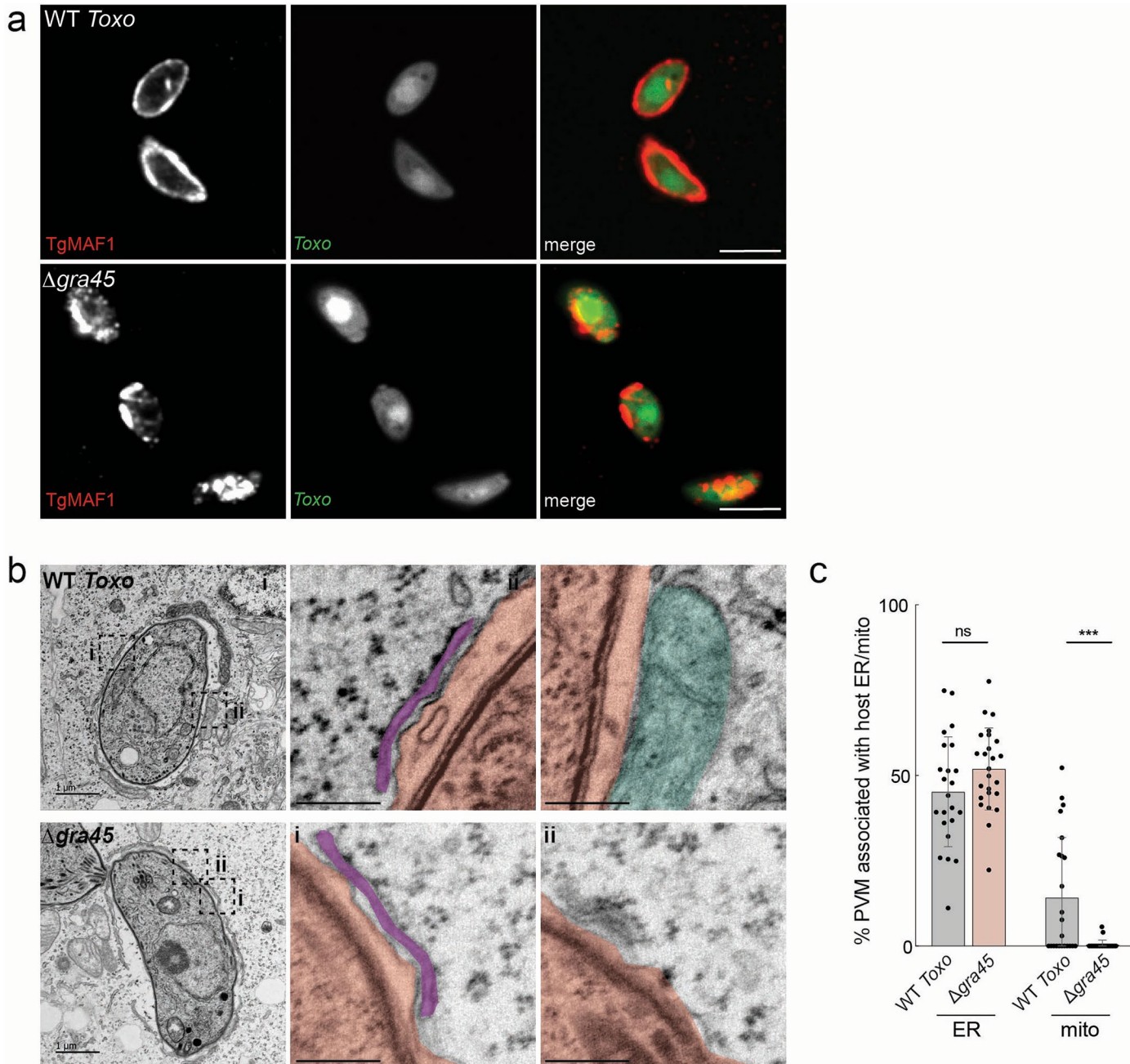

**Extended Data Fig. 5 | Host ER-*Toxoplasma* MCS form independently of *Toxoplasma* dense granule effectors. a**, Immunofluorescence image of ES-2 cells infected with WT or Δ*gra45* parasites at 3 h post infection. Scale bar: 5 µm. **b**, Representative electron microscopy images of ES-2s infected with WT (*Toxo*^mcherry) or Δ*gra45* parasites. MCS between the *Toxoplasma* parasite vacuole membrane (PVM) and (**i**) host ER and (**ii**) host mito. Data is representative of n = 1 biological replicate. Scale bars: 1 µm; inset, 250 nm. Pink, PV; purple, ER; turquoise, mito. **c**, Percentage of PVM associated with host ER and mitochondria in images as in (**b**). PVM: parasite vacuole membrane. EM data are mean ± SD from n = >20 vacuoles from 1 biological replicate (WT: 23; Δ*gra45*: 25 *Toxoplasma* vacuoles) ***p = 0,0003 by means of two-tailed unpaired t-test.

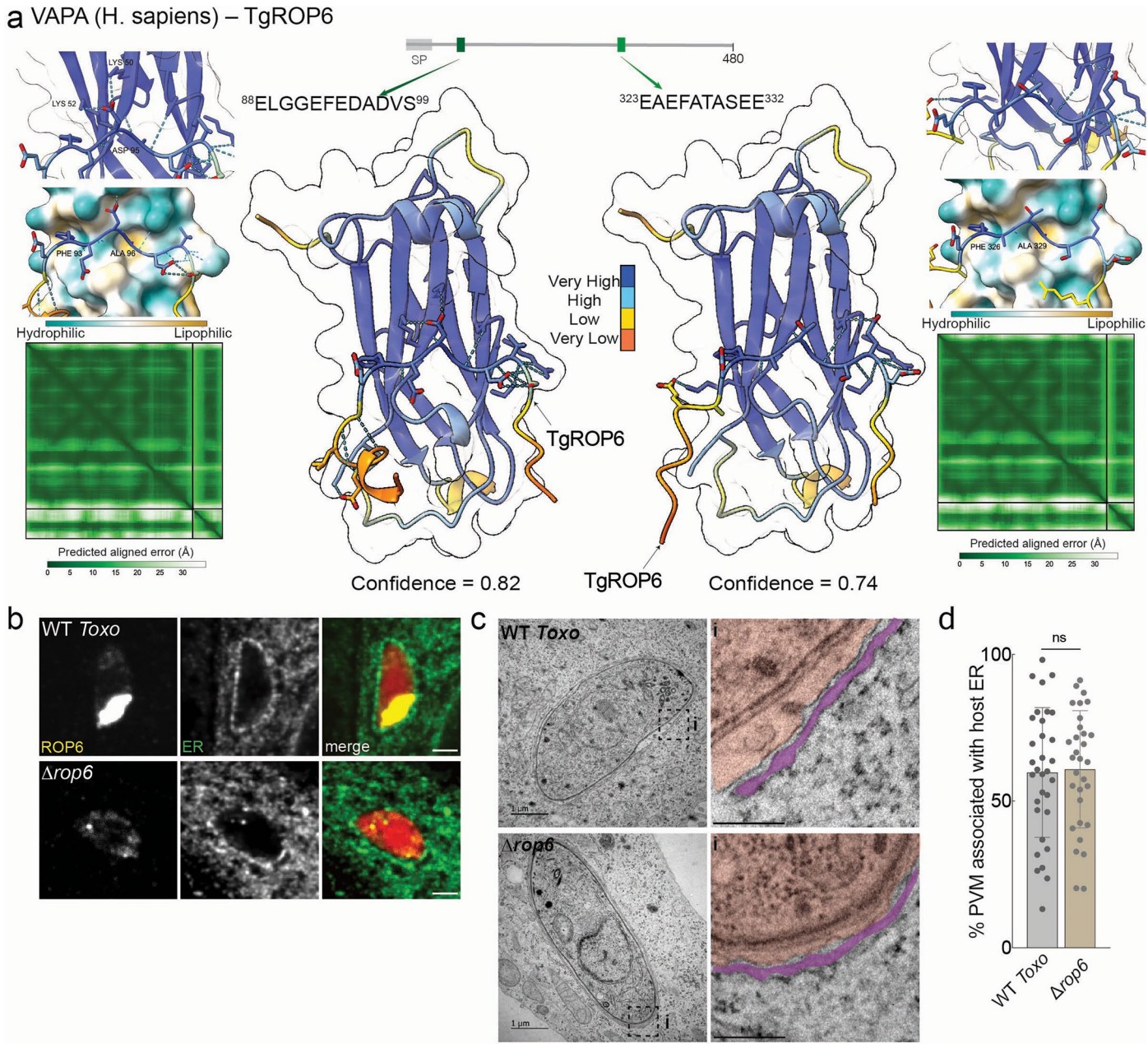

**Extended Data Fig. 6 | Loss of TgROP6 does not affect host ER-*Toxoplasma* MCS. a**, AlphaFold multimer models of the MSP domain of VAPA with the predicted canonical (Left; EFFDAxE) and modified (Right; ExFxDAxE) FFAT motifs of TgROP6. **b**, Representative immunofluorescence images of HFFs infected with WT (*Toxo*^PVMβ11) or Δ*rop6* parasites at 3 h post infection (hpi). Data is representative of n = 1 biological replicate. ER (calnexin). Scale bar: 2 μm. **c**, Representative electron microscopy images of HeLas infected with WT or

Δ*rop6* parasites at 3 hpi. Membrane contact sites between the *Toxoplasma* parasite vacuole membrane (PVM) and (**i**) host ER. Scale bars, 1 μm; inset, 250 nm. **d**, Percentage of PVM associated with host ER in images as in (**c**). Pink, parasite vacuole; purple, ER. EM data are mean ± SD from n > 30 vacuoles from 1 biological replicate (WT: 32; Δ*rop6*: 31 *Toxoplasma* vacuoles) by means of two-tailed unpaired t-test.

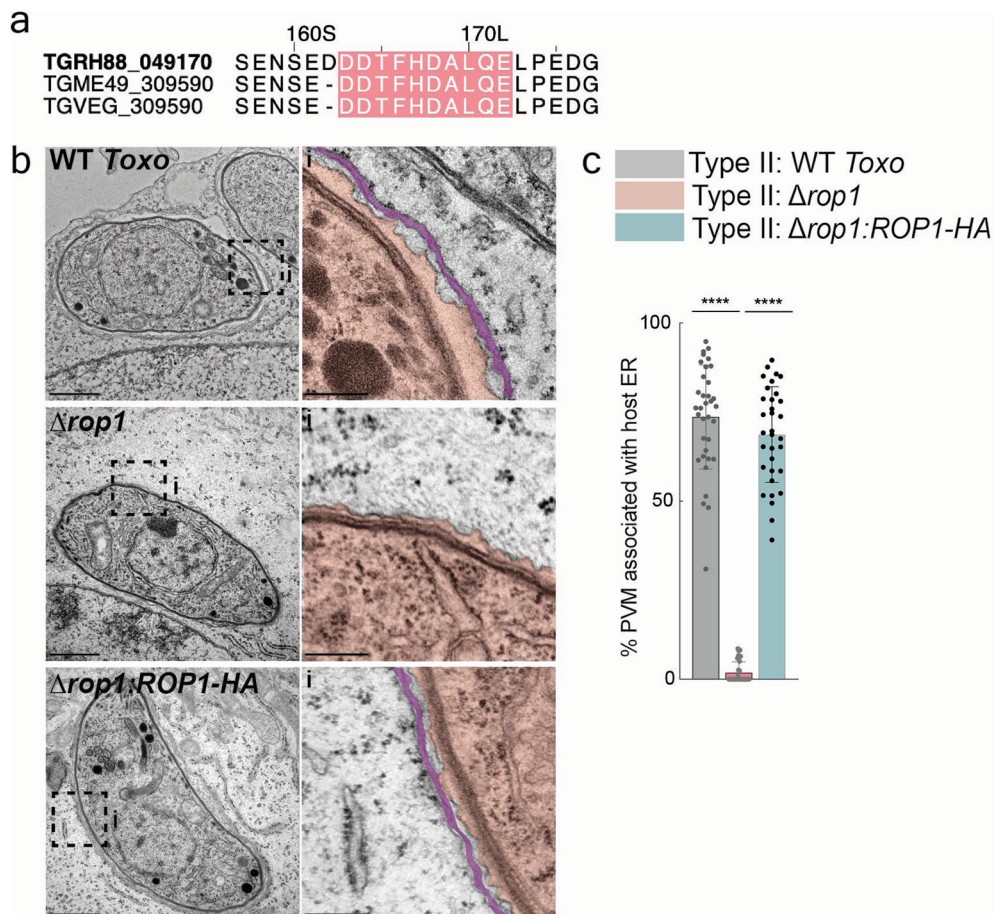

**Extended Data Fig. 7 | Type II TgROP1 mediate host ER-*Toxoplasma* MCS.**
**a**, Sequence alignment of TgROP1 orthologous sequences from the canonical
*Toxoplasma* strains- Type I (TGRH88_049170), Type II (TGME49_309590) and
Type III (TGVEG_309590); TgROP1 ExFxDAxE motif with high-confidence VAPA
interaction in pink. **b**, Representative electron micrograph images of ES-2s
infected with *Toxoplasma* WT, Δ*rop1*, and Δ*rop1:ROP1-HA* of the Type II lineage
at 3 h post infection. MCS between the *Toxoplasma* parasite vacuole membrane
(PVM) and (**i**) host ER. Scale bars: 1 μm; inset, 250 nm. Pink, PV; purple, ER.
**c**, Percentage of *Toxoplasma* PVM associated with host ER in images as in (**b**).
EM data are mean ± SD from n ≥30 vacuoles from 1 biological replicate (WT: 35;
Δ*rop1*:30; Δ*rop1:ROP1HA*: 33 *Toxoplasma* vacuoles). ****p < 0.0001 by means of
one-way ANOVA Tukey's multiple comparison test.

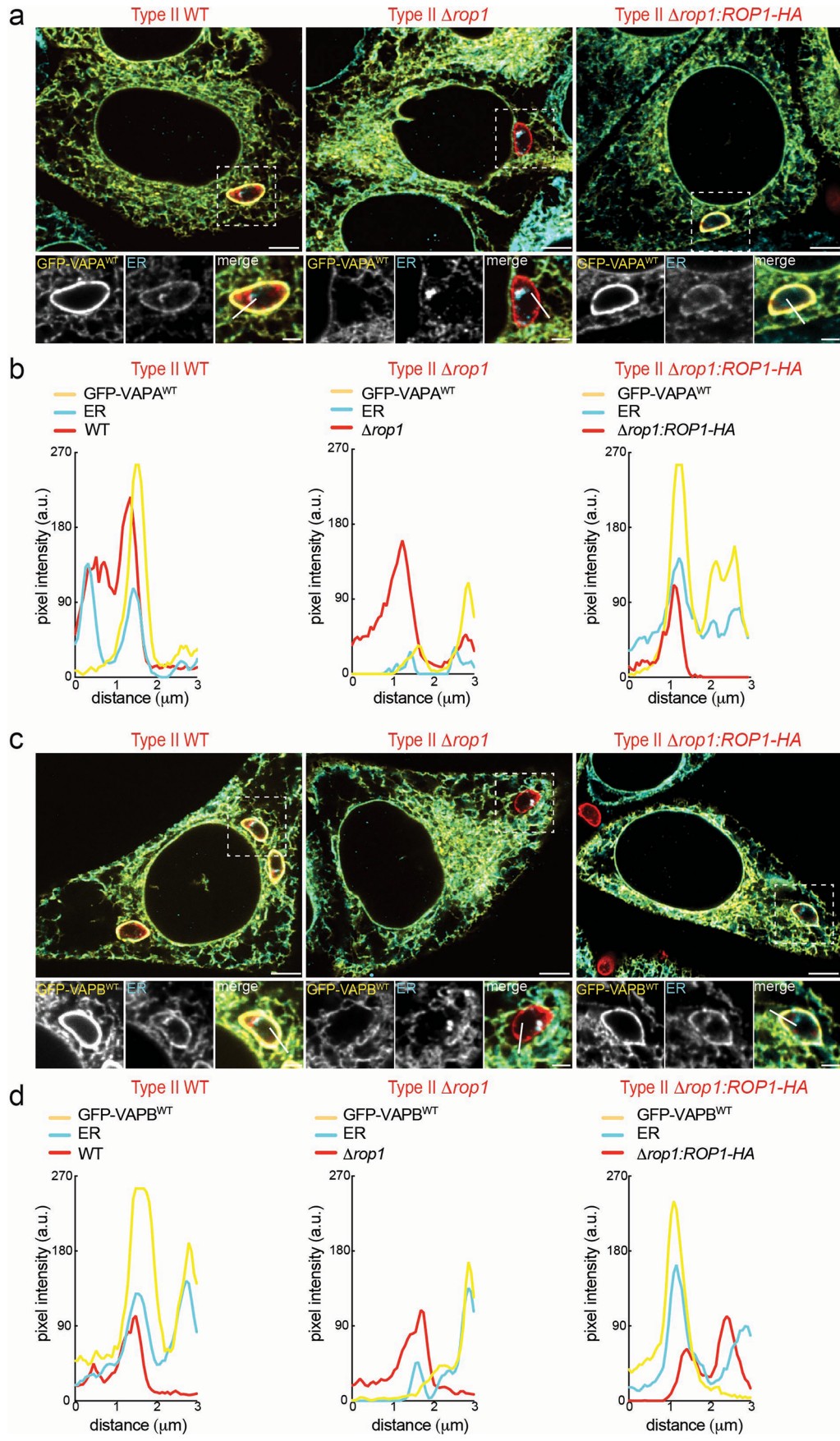

**Extended Data Fig. 8 | See next page for caption.**

**Extended Data Fig. 8 | VAPA and VAPB enrichment around Type II parasites requires TgROP1. a**, Representative immunofluorescence (IF) images of VAP DKO HeLa cells expressing GFP-VAPA^WT and infected with *Toxoplasma* Type II: WT, Δ*rop1*, Δ*rop1:ROP1-HA* parasites at 3 h post infection (hpi). *Toxoplasma* strains (surface antigen 1; TgSAG1); ER (calnexin). **b**, Corresponding pixel intensity plots for white line in (**a**) inset. **c**, Representative IF images of VAP DKO HeLa cells expressing GFP-VAPB^WT and infected with indicated parasites at 3 hpi. *Toxoplasma* strains (surface antigen 1; TgSAG1); ER (calnexin). **d**, Corresponding pixel intensity plots for white line in (**c**) inset. Scale bars: 5 μm; inset 2 μm. Data in all experiments is representative of n = 1 biological replicate.

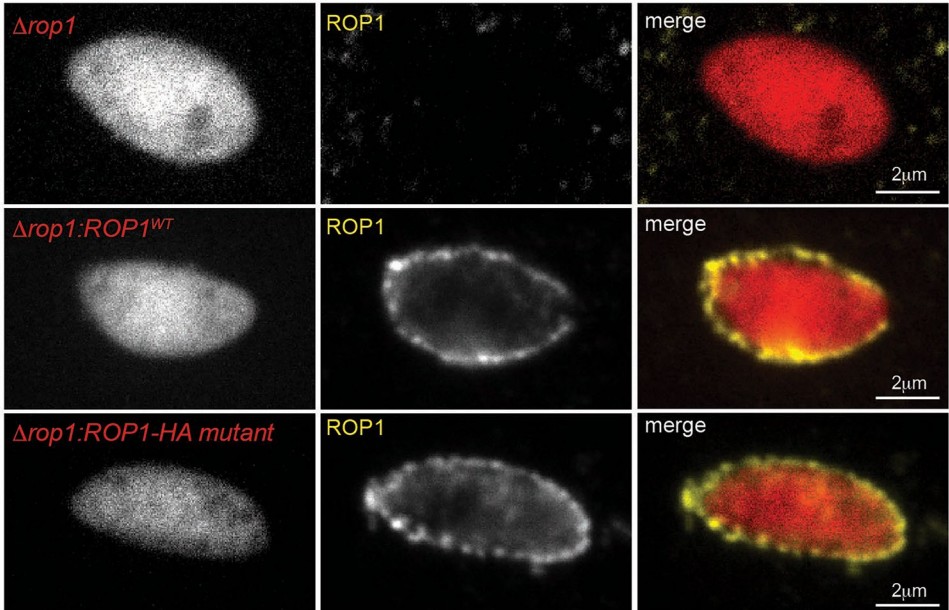

**Extended Data Fig. 9 | TgROP1F166A localizes to the PVM.** Representative immunofluorescence images of human foreskin fibroblasts infected with Type I: Δ*rop1*, Δ*rop1:ROP1^{WT}-HA*, and Δ*rop1:ROP1^{F166A}-HA* (mutant) parasites fixed at 3 h post infection (hpi) and permeabilized using 0.05% saponin. Scale bar: 2 μm. Data is representative of n = 1 biological replicate.

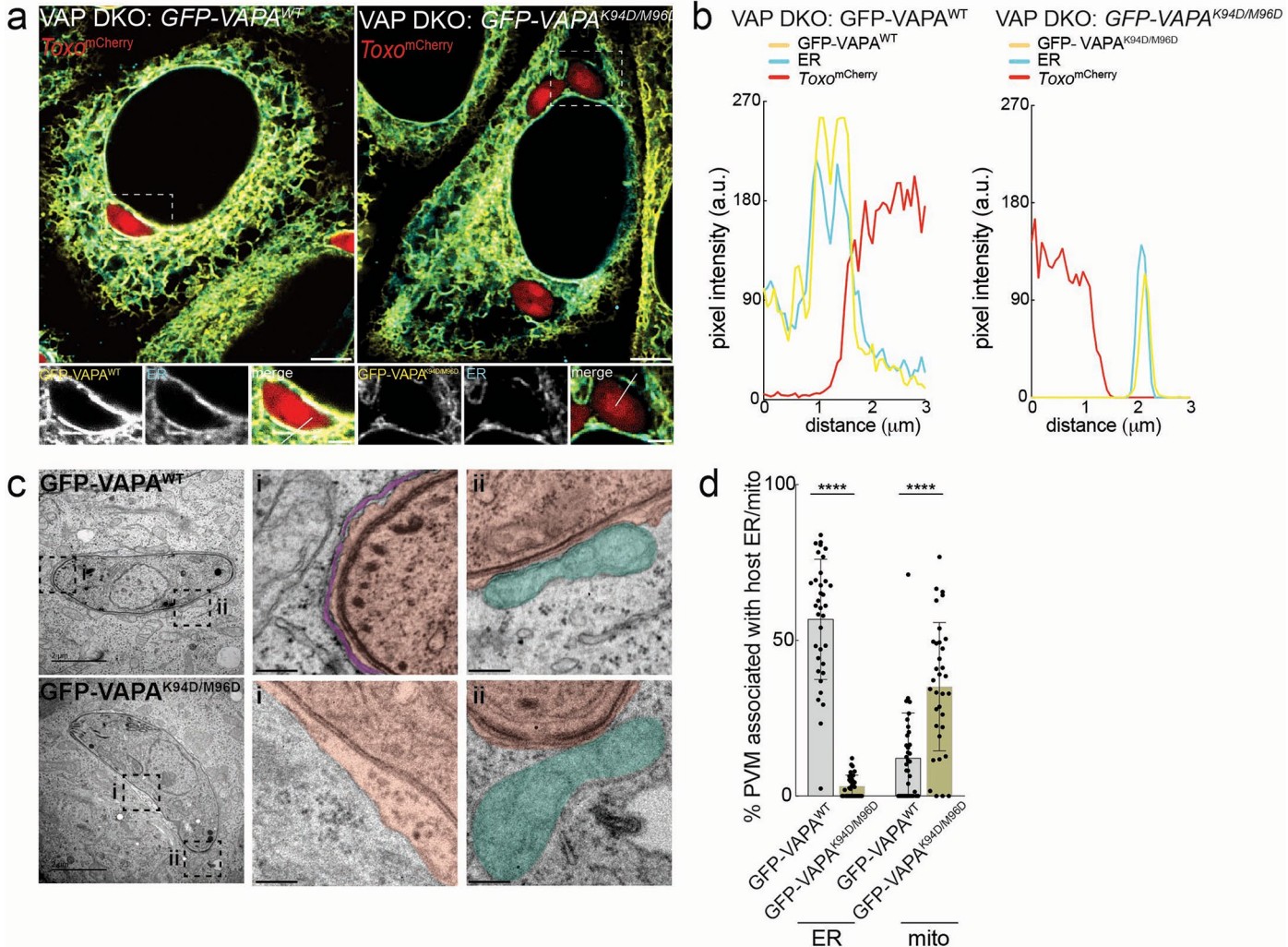

**Extended Data Fig. 10 | The VAPA MSP domain is critical for host ER-*Toxoplasma* MCS. a**, Representative immunofluorescence images of GFP-VAPA[WT] and GFP-VAPA[K94D/M96D] cells infected with *Toxo*[mCherry] at 3 h post infection (hpi). Data is representative of n = 2 biological replicates. ER (calnexin). Scale bars: 5 μm; inset, 2 μm. **b**, Corresponding pixel intensity plots for white line in (**a**) inset. **c**, Representative electron microscopy images of GFP-VAPA[WT] and GFP-VAPA[K94D/M96D] cells infected with *Toxo*[mCherry] at 3 hpi. Scale bars: 2 μm; inset, 250 nm. MCS between the *Toxoplasma* parasite vacuole membrane (PVM) and (**i**) host ER and (**ii**) host mito. Pink, PV; purple, ER; turquoise, mito. **d**, Percentage of *Toxoplasma* PVM associated with host ER and mitochondria in images as in (**c**) from n > 30 vacuoles from 1 biological replicate (GFP-VAPA[WT]: 35; GFP-VAPA[K94D/M96D]: 34 *Toxoplasma* vacuoles). ****p < 0.0001 by means of two-tailed unpaired t-test.

# Reporting Summary

## Statistics

For all statistical analyses, confirm that the following items are present in the figure legend, table legend, main text, or Methods section.

| n/a | Confirmed | |
|---|---|---|
| ☐ | ☒ | The exact sample size ($n$) for each experimental group/condition, given as a discrete number and unit of measurement |
| ☐ | ☒ | A statement on whether measurements were taken from distinct samples or whether the same sample was measured repeatedly |
| ☐ | ☒ | The statistical test(s) used AND whether they are one- or two-sided<br>*Only common tests should be described solely by name; describe more complex techniques in the Methods section.* |
| ☒ | ☐ | A description of all covariates tested |
| ☒ | ☐ | A description of any assumptions or corrections, such as tests of normality and adjustment for multiple comparisons |
| ☒ | ☐ | A full description of the statistical parameters including central tendency (e.g. means) or other basic estimates (e.g. regression coefficient) AND variation (e.g. standard deviation) or associated estimates of uncertainty (e.g. confidence intervals) |
| ☒ | ☐ | For null hypothesis testing, the test statistic (e.g. $F$, $t$, $r$) with confidence intervals, effect sizes, degrees of freedom and $P$ value noted<br>*Give P values as exact values whenever suitable.* |
| ☒ | ☐ | For Bayesian analysis, information on the choice of priors and Markov chain Monte Carlo settings |
| ☒ | ☐ | For hierarchical and complex designs, identification of the appropriate level for tests and full reporting of outcomes |
| ☒ | ☐ | Estimates of effect sizes (e.g. Cohen's $d$, Pearson's $r$), indicating how they were calculated |

*Our web collection on statistics for biologists contains articles on many of the points above.*

## Software and code

Policy information about availability of computer code

| Data collection | Data are available in supplementary tables. |
|---|---|
| Data analysis | A description of analytical methods provided either in the main manuscript or in materials and methods. |

For manuscripts utilizing custom algorithms or software that are central to the research but not yet described in published literature, software must be made available to editors and reviewers. We strongly encourage code deposition in a community repository (e.g. GitHub). See the Nature Portfolio guidelines for submitting code & software for further information.

## Data

Policy information about availability of data

All manuscripts must include a data availability statement. This statement should provide the following information, where applicable:

- Accession codes, unique identifiers, or web links for publicly available datasets
- A description of any restrictions on data availability
- For clinical datasets or third party data, please ensure that the statement adheres to our policy

Data are available in supplementary tables.

# Research involving human participants, their data, or biological material

Policy information about studies with [human participants or human data](). See also policy information about [sex, gender (identity/presentation), and sexual orientation]() and [race, ethnicity and racism]().

| | |
|---|---|
| Reporting on sex and gender | n/a |
| Reporting on race, ethnicity, or other socially relevant groupings | n/a |
| Population characteristics | n/a |
| Recruitment | n/a |
| Ethics oversight | n/a |

Note that full information on the approval of the study protocol must also be provided in the manuscript.

# Field-specific reporting

Please select the one below that is the best fit for your research. If you are not sure, read the appropriate sections before making your selection.

☒ Life sciences ☐ Behavioural & social sciences ☐ Ecological, evolutionary & environmental sciences

For a reference copy of the document with all sections, see [nature.com/documents/nr-reporting-summary-flat.pdf]()

# Life sciences study design

All studies must disclose on these points even when the disclosure is negative.

| | |
|---|---|
| Sample size | Sample size determination was based on prior studies (i.e. for FACS analyses). |
| Data exclusions | No exclusions. |
| Replication | Replicates indicated. |
| Randomization | n/a |
| Blinding | n/a |

# Reporting for specific materials, systems and methods

We require information from authors about some types of materials, experimental systems and methods used in many studies. Here, indicate whether each material, system or method listed is relevant to your study. If you are not sure if a list item applies to your research, read the appropriate section before selecting a response.

## Materials & experimental systems

| n/a | Involved in the study |
|---|---|
| ☐ | ☒ Antibodies |
| ☐ | ☒ Eukaryotic cell lines |
| ☒ | ☐ Palaeontology and archaeology |
| ☒ | ☐ Animals and other organisms |
| ☒ | ☐ Clinical data |
| ☒ | ☐ Dual use research of concern |
| ☒ | ☐ Plants |

## Methods

| n/a | Involved in the study |
|---|---|
| ☒ | ☐ ChIP-seq |
| ☐ | ☒ Flow cytometry |
| ☒ | ☐ MRI-based neuroimaging |

## Antibodies

| | |
|---|---|
| Antibodies used | For Immunofluorescence analyses: Primary Abs: Calnexin (GeneTex: GTX109669 [C3], C-term) or Calnexin (Proteintech:10427-2-AP); VAPA (Proteintech15275-1-AP); TOMM70 (HPA:048020); VAPB (Proteintech: 14477-1-AP); HA (CST #3724, Roche 3F10); c-Myc (CST: 5605S, D84C12); Antisera of TgMAF1 15; HA (CST:3724, Roche 3F10); GFP (Takara Bio: 632380); TgROP1 (Abnova: MAB17504); TgROP6 (mouse monoclonal) were used at 1:300 or 1:2000 O/N. Secondary Abs: Alexa Fluor Plus 405, Alexa Fluor Plus 488, Alexa |

Fluor Plus 594, Alexa Fluo Plus 647 (Thermo Fisher). For Immunoblotting: HMGCR (Sigma AMAB90619), TOMM70 (HPA048020); HA-HRP (Roche 12013819001); VAPA (Proteintech:15275-1-AP); VAPB (Proteintech: 14477-1-AP); Calnexin (Proteintech: 10427-2-AP); TgROP1(Abnova: MAB17504); TgROP6 (mouse monoclonal) and TgGra45 (Dr. D Soldati; U. of Geneva).

Validation | Validations were either indicated on relevant vendor website or validated in house (i.e. using knockout lines).

# Eukaryotic cell lines

Policy information about cell lines and Sex and Gender in Research

Cell line source(s) | HeLa adenocarcinoma cells, ES-2 ovary clear cell carcinoma and human foreskin fibroblasts (HFFs) cells were obtained from ATCC (CCL-2, CRL-1978, and SCRC-1041, respectively); VAP A/B double knockout (VAP DKO) cells were a kind gift from Dr. Pietro Di Camelli. All cells were cultured at 37°C and 5% CO2 in Dulbecco's Modified Eagle's GlutaMAXTM medium and supplemented with 10% heat-inactivated FBS (Gibco: A3840402) and 100 U/ml Penicillin-Streptomycin (Thermo Fisher Scientific: 15070063) (referred to as cDMEM). ROP1 knockout parasites from Moritz Treeck (reference in manuscript), Type III parasites were a gift from Martin Blume.

Authentication | VAPA/B DKOs and ROP1 / ROP6 KO parasite lines were validated using PCR and immunoblot / immunofluorescence analysis.

Mycoplasma contamination | Cells were routinely tested for Mycoplasma infection by polymerase chain reaction (PCR) every two weeks.

Commonly misidentified lines (See ICLAC register) | n/a

# Plants

Seed stocks | *Report on the source of all seed stocks or other plant material used. If applicable, state the seed stock centre and catalogue number. If plant specimens were collected from the field, describe the collection location, date and sampling procedures.*

Novel plant genotypes | *Describe the methods by which all novel plant genotypes were produced. This includes those generated by transgenic approaches, gene editing, chemical/radiation-based mutagenesis and hybridization. For transgenic lines, describe the transformation method, the number of independent lines analyzed and the generation upon which experiments were performed. For gene-edited lines, describe the editor used, the endogenous sequence targeted for editing, the targeting guide RNA sequence (if applicable) and how the editor was applied.*

Authentication | *Describe any authentication procedures for each seed stock used or novel genotype generated. Describe any experiments used to assess the effect of a mutation and, where applicable, how potential secondary effects (e.g. second site T-DNA insertions, mosiacism, off-target gene editing) were examined.*

# Flow Cytometry

## Plots

Confirm that:

☒ The axis labels state the marker and fluorochrome used (e.g. CD4-FITC).

☒ The axis scales are clearly visible. Include numbers along axes only for bottom left plot of group (a 'group' is an analysis of identical markers).

☐ All plots are contour plots with outliers or pseudocolor plots.

☐ A numerical value for number of cells or percentage (with statistics) is provided.

## Methodology

Sample preparation | For CRISPR screen: To perform the screen with technical duplicates, two vials of the split-GFP screen parasites (each considered as a technical duplicate) were thawed onto two T175 flasks of HFF monolayers. The next day the media of the flasks were changed to 25 μg/ml Mycophenolic acid and 50 μg/ml Xanthine (Sigma-Aldrich). Two days following treatment with selection media, the parasites were expanded by passing 2E6 parasites (to ensure a 1000x representation of guides) onto 6 T175 flasks of HFF monolayers. The next day, 300E6 OMM GFP1-10 ES-2 cells and ERM GFP1-10 ES-2 cells were plated in 15-cm dishes (8E6 cells/ dish). The next morning, split GFP-parasites from each technical replicate were used to infect 150E6 cells of each cell type at a low multiplicity of infection of 0.5 and left for 24 hours after infection. The following day, cells from each technical replicate (150 million cells) were trypsinized with accutase (to avoid clumpling) and pooled together into 50 ml falcons. The cells were fixed in 2% PFA for 5 minutes in FACS buffer with 5% accutase, spun down at 300 x g for 5 minutes to get rid of fixative. The cells were distributed into FACS tubes for sorting. The host mitochondria-Toxoplasma MCS screen cells were sorted using a BD FACSAria III sorter and the host ER-Toxoplasma MCS screen cells were sorted using a BD FACSFusion sorter. Gates were drawn to first sort for infected cells (mCherry fluorescence) and then all cells negative for GFP expression (GFPneg) and the top 20% of the GFP positive [GFP-high (GFPhi)] populations were sorted over the course of four days for both screens. Cell pellets were stored at -80°C; for growth analysis: For split GFP assays, monolayers of infected-ES2 or HeLa cells were rinsed with PBS, trypsinized and fixed in 2% paraformaldehyde in 3% FBS in 1XPBS (FACS buffer) for 5 min. After a spin at 300 rcf for 5 min, cells were resuspended in FACS buffer and 10,000 events were analyzed on a FACSFortessa using BD FACSDiva software for mCherry intensity and then tested for GFP expression. To assess parasite proliferation, monolayers of ES-2 cells infected with ToxomCherry parasites were left to grow for 24 hours post infection and harvested as

| | previously described18. 10,000 events were analyzed on a FACSFortessa and the mCherry median fluorescence intensity (mFI) using BD FACSDiva software. |
|---|---|
| Instrument | BD FACSAria III; BD FACSFusion; FACS Fortessa |
| Software | BD FACSDiva software |
| Cell population abundance | Gates were drawn to first sort for infected cells (mCherry fluorescence) and then all cells negative for GFP expression (GFPneg) and the top 20% of the GFP positive [GFP-high (GFPhi)] populations were sorted over the course of four days for both screens; purity was assessed by rerunning populations and assessing GFP expression |
| Gating strategy | Cells were initially gated using forward scatter (FSC) versus side scatter (SSC) to exclude debris. The resulting population was then analyzed based on mCherry intensity to distinguish between uninfected (mCherry-; UI) and infected (mCherry+; INF) cells. Subsequently, both the UI and infected INF were assessed for their GFP expression levels; GFP-negative samples were used to draw gates for GFP-positive cells. |

☒ Tick this box to confirm that a figure exemplifying the gating strategy is provided in the Supplementary Information.

