## [Peer Review File · Nature Microbiology]

Toxoplasma effector TgROP1 establishes membrane contact sites with the endoplasmic reticulum during infection

Corresponding Author: Professor Lena Pernas

Version 0:

Reviewer comments:

Reviewer #1

(Remarks to the Author)

Summary

In this manuscript an answer is given to the long-standing question of what mediates the *T. gondii* PVM association with the host cell's ER. The authors used a very elegant and innovative system for this, based on split GFP, CRISPR screens and FACS. First they show that they can use this to identify the PVM-mitochondria DCS factors, which is a good first step as it shows the split GFP on the PVM relies on dense granule and the GRA45 chaperone, which are 'false' hits in the ER screen. I also appreciate the detail on differences between Type I/III and Type II strains on how mitochondria-PVM contact affects the extend of ER-PVM contact. Ample of beautiful controls throughout the work, but the inability for the split GFP itself to drive DCS took the cake. The rationalization and prioritization of the Rhoptry candidates, based on the insight of the FFAT motif and likely candidacy of VAPA/B on the ER site is very well presented and pursued, and then experimentally puts ROP1 center stage with a non-essential function for ROP6. Then the exact nature of the interaction domains is mapped, and the insight is used to demonstrate a growth advantage for ER-PVM contact, which adds biological rigor and significance, respectively. The rationale and discussion often refer to Chlamydia for context, and point to lipid access as the potential role. This is very logical, but the one experiment done in this direction (Fig 5g) does not add much to the story. The title is also wearing a large size pants, which actually does not capture the story very well.

Specific points

1. The title is quite abstract and too large for what is reported here: the identification of the factors mediating PVM-ER contact by a forward genetic screen. Since this SPLICS system was already used before, although not in a trans-kingdom context, it does not carry sufficient novelty to justify the title.
2. Fig 6g (& discussion lines 376-387). This is a cool experiment, but without validation that lipid access/transfer is the mechanism, this is a bit of a distraction from the rest of this great study. Together with the discussion offering several scenarios that cannot be differentiated, my suggestion is to take it out of this manuscript to stay focused.
3. Fig 6h. The growth deficiency in absence of the ER-PVM DCS. Only 1 time point at 24 hpi is shown. Only mean FI is plotted, suggesting lower parasite burden. However, this lower burden can be due to a delayed invasion efficiency as well as slower replication, which cannot be discerned in the way the assay is set up. The flow data should contain the number of infected cells, which if the same between conditions, would hint that delayed invasion is not a problem. However, and additional time point, e.g. 12 hrs for both mean FI and % infected cells would consolidate this even more.

Minor points

1. Fig 2a. The font in the constructs is too small to read without magnification
2. Line 172/3. Fig 2d should be 2d here

Reviewer #2

(Remarks to the Author)

The work describe in this manuscript addresses an important and timely topic by asking to study membrane contact sites (MCS) that likely contribute to host parasite interaction. The authors first established the presence of ER-parasite-vacuole MCS in the three major strains of the intracellular parasite *Toxoplasma gondii* and characterise the properties of those contacts. The authors then attempted to combine a new sensor of host-organelle-parasite-vacuole MCS with a loss-of-function CRISPR screening to discover an ER-parasite-vacuole molecular tethers. This led to the identification of molecules that are important for ER recruitment to the parasite vacuoles.

There are a few major weaknesses.

Sensor - the development of a sensor to study host-Toxoplasma interaction has potential to be valuable to study this and other intracellular pathogens once fully validated, however better controls are required for the split GFP sensor presented. While the

TOM70KO experiment provides some evidence for a dynamic response of the sensor to cellular defects, it doesn't justify the conclusion that "Thus, our host-pathogen MCS sensor recapitulates Toxoplasma-mitochondria contact site biology" as its potential effect of artificial MCS could not be excluded with the current control. A more relevant control is to compare the mitochondria-vacuole MCS with and without the sensor via in the same experiments used to define and characterise MSC in the first paragraph. It is further important to evaluate the effect of the sensor on the parasite survivability over time. Finally, a specific control should address the relevant MCS being studied later in the work. In other systems, such as yeast, abundant membrane proteins on each studied membrane fused to a reporter have been successfully used as sensors for MCS and the same could be applied here.

Candidate selection - the selection of parasite rhopty proteins as potential tethers included some biased decisions or application of arbitrary cutoffs. For example, since Gra45 had no effect on MCS the authors conclude that Gra proteins are identified due to their role in vacuole biology and insertion of the Maf protein, then continue to focus on Rop proteins. However, the same logic can apply for Rop – namely that their effect may be due to a role in vacuole biology rather than a tether role. Likewise, the motif used to identify VAP-interacting proteins, was arbitrarily "relaxed" to identify more candidates. Similarly, the pLDDT cutoff is arbitrary. The question is then, was the screen useful or did the authors ended up taking an educated guess. What would be the outcome if all known PMV proteins were analysed in the same way. The readers should be able to appreciate if the screen provided any added value.

Rop1-VAP interactions – the IP experiment described in lines 308-315 lacks controls for specificity of the interaction VAPA/B (given that they are abundant proteins on membranes that interact). There may well be more un-specific interaction with abundant ER proteins and only one ER marker was used to exclude this possibility. IP with Rop1 followed by MS is more suitable.

While the manuscript described several elegant but indirect experiment to support a potential Rop1-VAPA/B tether for host-ER-parasite-vacuole MCS (such Rop 1 dependent recruitment of VAP; MCS dependency on VAP). The work lack an experiment that demonstrate this directly. At the same time the indirect experiments lack critical controls. For example, the authors show VAP DKO results in loss of ER-vacuole association however is this specific or perhaps a general ER defect? What is the effect on other ER MCS (e.g. host ER-mitochondria MCS)? What is the effect on ER biogenesis and function?

Reviewer #3

(Remarks to the Author)

Interorganellar membrane contact sites (which the authors call MCS) have radically altered our understanding of organellar function. Unsurprisingly, intracellular pathogens manipulate MCS in order to exploit their host cell's physiology. It is well-established that Toxoplasma has contacts with both the host mitochondria and ER. In the present study, the authors build upon some of their previous work in characterizing the Toxoplasma PVM-mitochondria to examine the PVM contacts with the host ER. Using an innovative biosensor and candidate-based CRISPR screen, the authors identify an intriguing parasite effector candidate in ROP1. They demonstrate convincingly that ROP1 expression is required for ER-PVM contacts, and that host VAPA/B are required on the host side.

Overall, the experiments are well described and the data are of high quality. I'm very excited by the potential here, but what strongly diminishes my excitement is concern over some sort of pleiotropic effect. Briefly, as the authors point out, GRA45 was originally identified in a very similar screen looking for proteins that altered host immune signaling, but has nothing (directly) to do with that; it appears to act far upstream, somewhere in trafficking of the proteins themselves. In the original paper describing rescue of the ROP1 knockout (pmid 8719248), the rhoptries were reported to be grossly changed in their morphology, a phenotype that was rescued by reintroduction of ROP1 expression (this was confirmed in the Butterworth et al paper from one of the co-authors labs). It is therefore possible that ROP1, like GRA45, has some sort of upstream effect that is overwhelming the screen. That ROP1 (like GRA45) has come up in multiple CRISPR screens for effectors is further cause for concern. Unfortunately, the experiments the authors have performed do not rule out such an explanation. With a few more reasonably straightforward experiments (described below), though, I'd be quite convinced.

- Other proteins appear de-enriched to at least the extent that ROP1 is in the mutant vs. wt VAPA IP (Fig 5b). Can the authors comment on any of these other proteins?

- In my opinion, to really nail that ROP1 is directly binding VAPA/B, the authors need to make some sort of point mutant(s) (not deletions of residues!) that do not rescue the ER-PVM MCS when expressed in Δ rop1. This should be reasonably straightforward given the AlphaFold predictions.

- Given that both ROP1 and VAPA can easily be expressed in bacteria, the authors could further strengthen their claims of direct binding by e.g. GST-pulldown. This would also allow them an excellent platform to demonstrate/screen the relevance of any mutants.

- Another concern – given that overexpression of the mutants in VAP-A that the authors use in their study have been demonstrated to induce gross changes in ER morphology (e.g. pmid 16004875), is it possible that the effects they see in their knockout or mutants of VAP-A are also potentially secondary? Since that came from a (very reasonable) hypothesis rather than a screen, extra controls may be warranted.

- It seems a bit odd that almost all of the experiments are performed very early after infection (~3 hours; before any replication). Does the nature of ER contacts change after the parasite has replicated 1 or more times? Can the authors add some description

of their reasoning for this?

Minor:

- First off – what is the provenance of the Δ rop1 and complement strain? Its generation is not described in the methods (just Δ rop6) and it is not cited as having been received from another source (nor are references for it cited – I'm assuming the recent strains made in the Treeck lab). [[ok I re-read and found the Butterworth reference here, but it was not entirely clear to me that this was the source of the relevant parasites – the source of all strains should be made clear in the methods as well]]

- I appreciate the attention to detail in the Methods. (other than neglecting to mention where the Δ rop1 came from)

- Many of the color choices used to annotate images are hard to read on the black backgrounds, and in some cases (especially blues) are not color blind friendly. A white stroke would help with this (and/or choosing slightly brighter colors).

- In all tables featuring gene models, it would be helpful to the reader to include gene names/predictions where available (e.g. proteomics tables)

Decision Letter:

6th February 2025

Dear Lena,

Thank you for your patience while your manuscript "An organelle-pathogen sensor reveals factors required for trans-kingdom contact sites" was under peer-review at Nature Microbiology. It has now been seen by 3 referees, whose expertise and comments you will find at the end of this email. Although they find your work of some potential interest, they have raised a number of concerns that will need to be addressed before we can consider publication of the work in Nature Microbiology.

In particular, while the referees found the study interesting, and referee #1 and #2 were more positive, all referees request additional control experiments, and have some concerns over indirect and potentially pleiotropic effects. Referee #1 suggests to add a 12h timepoint for the growth deficiency experiment in absence of the ER-PVM DCS. Referee #2 says that better controls are required for the split GFP sensor, and that a more relevant control would be to compare the mitochondria-vacuole MCS with and without the sensor. Furthermore, this referee says that it will be important to evaluate the effect of the sensor on the parasite survivability over time, and that a specific control should address the relevant MCS being studied later in the work. Referee #2 also states that the pLDDT cutoff is arbitrary, and that the IP experiment lacks controls for specificity of the interaction. The referee notes that IP with Rop1 followed by MS would be more suitable. Further, the referee asks for more direct experiments to support a potential Rop1-VAPA/B tether for host-ER-parasite-vacuole, and/or to better control the indirect experiments. Referee #3 has concerns over potential pleiotropic effects, and says it could be possible that "ROP1, like GRA45, has some sort of upstream effect that is overwhelming the screen." The referee suggests adding a point mutant(s) (not deletions of residues) that do not rescue the ER-PVM MCS when expressed in Δ rop1, and to perform GST-pulldown to strengthen the claims of direct binding. This referee is also concerned that the effects in the knockout or mutants of VAP-A may also be potentially secondary, and says that additional controls may be required. Editorially, we will need you to address the referee comments in full through additional experiments.

Should further experimental data allow you to address these criticisms, we would be happy to look at a revised manuscript.

Please include a data availability statement as a separate section after Methods but before references, under the heading "Data Availability". This section should inform readers about the availability of the data used to support the conclusions of your study. This information includes accession codes to public repositories (data banks for protein, DNA or RNA sequences, microarray, proteomics data etc...), references to source data published alongside the paper, unique identifiers such as URLs to data repository entries, or data set DOIs, and any other statement about data availability. At a minimum, you should include the following statement: "The data that support the findings of this study are available from the corresponding author upon request", mentioning any restrictions on availability. If DOIs are provided, we also strongly encourage including these in the Reference list (authors, title, publisher (repository name), identifier, year). For more guidance on how to write this section please see: <http://www.nature.com/authors/policies/data/data-availability-statements-data-citations.pdf>

* Include a "Response to referees" document detailing, point-by-point, how you addressed each referee comment. If no action

was taken to address a point, you must provide a compelling argument. This response will be sent back to the referees along with the revised manuscript.

* If you have not done so already we suggest that you begin to revise your manuscript so that it conforms to our Article format instructions at <http://www.nature.com/nmicrobiol/info/final-submission>. Refer also to any guidelines provided in this letter.

When submitting the revised version of your manuscript, please pay close attention to our [href="https://www.nature.com/nature-portfolio/editorial-policies/image-integrity">Digital Image Integrity Guidelines.](https://www.nature.com/nature-portfolio/editorial-policies/image-integrity) and to the following points below:

EXTENDED DATA FIGURES

Link Redacted

Note: This url links to your confidential homepage and associated information about manuscripts you may have submitted or be reviewing for us. If you wish to forward this e-mail to co-authors, please delete this link to your homepage first.

Nature Microbiology is committed to improving transparency in authorship. As part of our efforts in this direction, we are now requesting that all authors identified as 'corresponding author' on published papers create and link their Open Researcher and Contributor Identifier (ORCID) with their account on the Manuscript Tracking System (MTS), prior to acceptance. This applies to primary research papers only. ORCID helps the scientific community achieve unambiguous attribution of all scholarly contributions. You can create and link your ORCID from the home page of the MTS by clicking on 'Modify my Springer Nature account'. For more information please visit [please visit www.springernature.com/orcid](http://www.springernature.com/orcid).

If you wish to submit a suitably revised manuscript we would hope to receive it within 6 months. If you cannot send it within this time, please let us know.

Yours sincerely,

Reviewer Expertise:

Referee #1: Toxoplasma pathogen-host interaction, proteomics

Referee #2: Toxoplasma biology, membrane contact sites

Referee #3: Electron microscopy, Toxoplasma

Reviewer Comments:

Reviewer #1 (Remarks to the Author):

Summary

In this manuscript an answer is given to the long-standing question of what mediates the *T. gondii* PVM association with the host cell's ER. The authors used a very elegant and innovative system for this, based on split GFP, CRISPR screens and FACS. First they show that they can use this to identify the PVM-mitochondria DCS factors, which is a good first step as it shows the split GFP on the PVM relies on dense granule and the GRA45 chaperone, which are 'false' hits in the ER screen. I also appreciate the detail on differences between Type I/III and Type II strains on how mitochondria-PVM contact affects the extend of ER-PVM contact. Ample of beautiful controls throughout the work, but the inability for the split GFP itself to drive DCS took the cake. The rationalization and prioritization of the Rhopty candidates, based on the insight of the FFAT motif and likely candidacy of VAPA/B on the ER site is very well presented and pursued, and then experimentally puts ROP1 center stage with a non-

essential function for ROP6. Then the exact nature of the interaction domains is mapped, and the insight is used to demonstrate a growth advantage for ER-PVM contact, which adds biological rigor and significance, respectively. The rationale and discussion often refer to Chlamydia for context, and point to lipid access as the potential role. This is very logical, but the one experiment done in this direction (Fig 5g) does not add much to the story. The title is also wearing a large size pants, which actually does not capture the story very well.

Specific points

1. The title is quite abstract and too large for what is reported here: the identification of the factors mediating PVM-ER contact by a forward genetic screen. Since this SPLICS system was already used before, although not in a trans-kingdom context, it does not carry sufficient novelty to justify the title.
2. Fig 6g (& discussion lines 376-387). This is a cool experiment, but without validation that lipid access/transfer is the mechanism, this is a bit of a distraction from the rest of this great study. Together with the discussion offering several scenarios that cannot be differentiated, my suggestion is to take it out of this manuscript to stay focused.
3. Fig 6h. The growth deficiency in absence of the ER-PVM DCS. Only 1 time point at 24 hpi is shown. Only mean FI is plotted, suggesting lower parasite burden. However, this lower burden can be due to a delayed invasion efficiency as well as slower replication, which cannot be discerned in the way the assay is set up. The flow data should contain the number of infected cells, which if the same between conditions, would hint that delayed invasion is not a problem. However, an additional time point, e.g. 12 hrs for both mean FI and % infected cells would consolidate this even more.

Minor points

1. Fig 2a. The font in the constructs is too small to read without magnification
2. Line 172/3. Fig 2d should be 2d here

Reviewer #2 (Remarks to the Author):

The work describe in this manuscript addresses an important and timely topic by asking to study membrane contact sites (MCS) that likely contribute to host parasite interaction. The authors first established the presence of ER-parasite-vacuole MCS in the three major strains of the intracellular parasite *Toxoplasma gondii* and characterise the properties of those contacts. The authors then attempted to combine a new sensor of host-organelle-parasite-vacuole MCS with a loss-of-function CRISPR screening to discover an ER-parasite-vacuole molecular tethers. This led to the identification of molecules that are important for ER recruitment to the parasite vacuoles.

There are a few major weaknesses.

Sensor - the development of a sensor to study host-*Toxoplasma* interaction has potential to be valuable to study this and other intracellular pathogens once fully validated, however better controls are required for the split GFP sensor presented. While the TOM70KO experiment provides some evidence for a dynamic response of the sensor to cellular defects, it doesn't justify the conclusion that "Thus, our host-pathogen MCS sensor recapitulates *Toxoplasma*-mitochondria contact site biology" as its potential effect of artificial MCS could not be excluded with the current control. A more relevant control is to compare the mitochondria-vacuole MCS with and without the sensor via in the same experiments used to define and characterise MSC in the first paragraph. It is further important to evaluate the effect of the sensor on the parasite survivability over time. Finally, a specific control should address the relevant MCS being studied later in the work. In other systems, such as yeast, abundant membrane proteins on each studied membrane fused to a reporter have been successfully used as sensors for MCS and the same could be applied here.

Candidate selection - the selection of parasite rhopty proteins as potential tethers included some biased decisions or application of arbitrary cutoffs. For example, since Gra45 had no effect on MCS the authors conclude that Gra proteins are identified due to their role in vacuole biology and insertion of the Maf protein, then continue to focus on Rop proteins. However, the same logic can apply for Rop – namely that their effect may be due to a role in vacuole biology rather than a tether role. Likewise, the motif used to identify VAP-interacting proteins, was arbitrarily "relaxed" to identify more candidates. Similarly, the pLDDT cutoff is arbitrary. The question is then, was the screen useful or did the authors ended up taking an educated guess. What would be the outcome if all known PMV proteins were analysed in the same way. The readers should be able to appreciate if the screen provided any added value.

Rop1-VAP interactions – the IP experiment described in lines 308-315 lacks controls for specificity of the interaction VAPA/B (given that they are abundant proteins on membranes that interact). There may well be more un-specific interaction with abundant ER proteins and only one ER marker was used to exclude this possibility. IP with Rop1 followed by MS is more suitable.

While the manuscript described several elegant but indirect experiment to support a potential Rop1-VAPA/B tether for host-ER-parasite-vacuole MCS (such Rop 1 dependent recruitment of VAP; MCS dependency on VAP). The work lack an experiment that demonstrate this directly. At the same time the indirect experiments lack critical controls. For example, the authors show VAP DKO results in loss of ER-vacuole association however is this specific or perhaps a general ER defect? What is the effect on other ER MCS (e.g. host ER-mitochondria MCS)? What is the effect on ER biogenesis and function?

Reviewer #3 (Remarks to the Author):

Interorganellar membrane contact sites (which the authors call MCS) have radically altered our understanding of organellar function. Unsurprisingly, intracellular pathogens manipulate MCS in order to exploit their host cell's physiology. It is well-established that *Toxoplasma* has contacts with both the host mitochondria and ER. In the present study, the authors build upon some of their previous work in characterizing the *Toxoplasma* PVM-mitochondria to examine the PVM contacts with the host ER. Using an innovative biosensor and candidate-based CRISPR screen, the authors identify an intriguing parasite effector candidate in ROP1. They demonstrate convincingly that ROP1 expression is required for ER-PVM contacts, and that host VAPA/B are required on the host side.

Overall, the experiments are well described and the data are of high quality. I'm very excited by the potential here, but what strongly diminishes my excitement is concern over some sort of pleiotropic effect. Briefly, as the authors point out, GRA45 was originally identified in a very similar screen looking for proteins that altered host immune signaling, but has nothing (directly) to do with that; it appears to act far upstream, somewhere in trafficking of the proteins themselves. In the original paper describing rescue of the ROP1 knockout (pmid 8719248), the rhoptries were reported to be grossly changed in their morphology, a phenotype that was rescued by reintroduction of ROP1 expression (this was confirmed in the Butterworth et al paper from one of the co-authors labs). It is therefore possible that ROP1, like GRA45, has some sort of upstream effect that is overwhelming the screen. That ROP1 (like GRA45) has come up in multiple CRISPR screens for effectors is further cause for concern. Unfortunately, the experiments the authors have performed do not rule out such an explanation. With a few more reasonably straightforward experiments (described below), though, I'd be quite convinced.

- Other proteins appear de-enriched to at least the extent that ROP1 is in the mutant vs. wt VAPA IP (Fig 5b). Can the authors comment on any of these other proteins?
- In my opinion, to really nail that ROP1 is directly binding VAPA/B, the authors need to make some sort of point mutant(s) (not deletions of residues!) that do not rescue the ER-PVM MCS when expressed in Δ rop1. This should be reasonably straightforward given the AlphaFold predictions.
- Given that both ROP1 and VAPA can easily be expressed in bacteria, the authors could further strengthen their claims of direct binding by e.g. GST-pulldown. This would also allow them an excellent platform to demonstrate/screen the relevance of any mutants.
- Another concern – given that overexpression of the mutants in VAP-A that the authors use in their study have been demonstrated to induce gross changes in ER morphology (e.g. pmid 16004875), is it possible that the effects they see in their knockout or mutants of VAP-A are also potentially secondary? Since that came from a (very reasonable) hypothesis rather than a screen, extra controls may be warranted.
- It seems a bit odd that almost all of the experiments are performed very early after infection (~3 hours; before any replication). Does the nature of ER contacts change after the parasite has replicated 1 or more times? Can the authors add some description of their reasoning for this?
Minor:
 - First off – what is the provenance of the Δ rop1 and complement strain? Its generation is not described in the methods (just Δ rop6) and it is not cited as having been received from another source (nor are references for it cited – I'm assuming the recent strains made in the Trecek lab). [[ok I re-read and found the Butterworth reference here, but it was not entirely clear to me that this was the source of the relevant parasites – the source of all strains should be made clear in the methods as well]]
- I appreciate the attention to detail in the Methods. (other than neglecting to mention where the Δ rop1 came from)
- Many of the color choices used to annotate images are hard to read on the black backgrounds, and in some cases (especially blues) are not color blind friendly. A white stroke would help with this (and/or choosing slightly brighter colors).
- In all tables featuring gene models, it would be helpful to the reader to include gene names/predictions where available (e.g. proteomics tables)

Version 1:

Reviewer comments:

Reviewer #1

(Remarks to the Author)

This revised manuscript effectively addresses my raised concerns, whereas several extra experiments suggested raise its profile even further. In particular exploiting the differences between *T. gondii* genotype strains together with the FFAT point mutant consolidates the conclusions. No further comments.

Reviewer #3

(Remarks to the Author)

The authors have adequately addressed my previous concerns.

Decision Letter:

Our ref: NMICROBIOL-24113714A

18th September 2025

Dear Lena,

Thank you for submitting your revised manuscript "Identification of host ER-Toxoplasma contact site tethers" (NMICROBIOL-24113714A). It has now been seen by two of the three original referees and their comments are below. The reviewers find that the paper has improved in revision, and therefore we'll be happy in principle to publish it in Nature Microbiology, pending minor revisions to comply with our editorial and formatting guidelines.

We are now performing detailed checks on your paper and will send you a checklist detailing our editorial and formatting requirements in about two weeks. Please do not upload the final materials and make any revisions until you receive this additional information from us.

Thank you again for your interest in Nature Microbiology. Please do not hesitate to contact me if you have any questions.

Sincerely,

Reviewer #1 (Remarks to the Author):

This revised manuscript effectively addresses my raised concerns, whereas several extra experiments suggested raise its profile even further. In particular exploiting the differences between *T. gondii* genotype strains together with the FFAT point mutant consolidates the conclusions. No further comments.

Reviewer #3 (Remarks to the Author):

The authors have adequately addressed my previous concerns.

Version 2:

Decision Letter:

15th October 2025

Dear Lena,

I am pleased to accept your Article "Toxoplasma effector TgROP1 establishes membrane contact sites with the endoplasmic reticulum during infection" for publication in Nature Microbiology. Thank you for having chosen to submit your work to us and many congratulations.

After the grant of rights is completed, you will receive a link to your electronic proof via email with a request to make any corrections within 48 hours. If, when you receive your proof, you cannot meet this deadline, please inform us at rjsproduction@springernature.com immediately. You will not receive your proofs until the publishing agreement has been

received through our system

Authors may need to take specific actions to achieve compliance with funder and institutional open access mandates. If your research is supported by a funder that requires immediate open access (e.g. according to [Plan S principles](https://www.springernature.com/gp/open-science/plan-s-compliance) or the [NIH public access policy](https://www.springernature.com/gp/open-science/us-federal-agency-compliance)) then you should select the gold OA route, and we will direct you to the compliant route where possible. Because authors warrant under our subscription licensing terms that they haven't committed to licensing any version of their article under a licence inconsistent with the terms of our agreement – including the applicable embargo period – publication under the subscription model isn't suitable for authors whose funders require no embargo.

Congratulations once again and I look forward to seeing the article published.

With kind regards,

P.S. Click on the following link if you would like to recommend Nature Microbiology to your librarian <http://www.nature.com/subscriptions/recommend.html#forms>

** Visit the Springer Nature Editorial and Publishing website at http://editorial-jobs.springernature.com?utm_source=ejP_NMicro_email&utm_medium=ejP_NMicro_email&utm_campaign=ejP_NMicro for more information about our career opportunities. If you have any questions please click [here](mailto:editorial.publishing.jobs@springernature.com).

[Reviewers comments are in bold; our responses are in normal font]

Reviewer Comments:

Reviewer #1 (Remarks to the Author):

Summary

In this manuscript an answer is given to the long-standing question of what mediates the *T. gondii* PVM association with the host cell's ER. The authors used a very elegant and innovative system for this, based on split GFP, CRISPR screens and FACS. First they show that they can use this to identify the PVM-mitochondria DCS factors, which is a good first step as it shows the split GFP on the PVM relies on dense granule and the GRA45 chaperone, which are 'false' hits in the ER screen. I also appreciate the detail on differences between Type I/III and Type II strains on how mitochondria-PVM contact affects the extend of ER-PVM contact. Ample of beautiful controls throughout the work, but the inability for the split GFP itself to drive DCS took the cake. The rationalization and prioritization of the Rhoptyr candidates, based on the insight of the FFAT motif and likely candidacy of VAPA/B on the ER site is very well presented and pursued, and then experimentally puts ROP1 center stage with a non-essential function for ROP6. Then the exact nature of the interaction domains is mapped, and the insight is used to demonstrate a growth advantage for ER-PVM contact, which adds biological rigor and significance, respectively. The rationale and discussion often refer to Chlamydia for context, and point to lipid access as the potential role. This is very logical, but the one experiment done in this direction (Fig 5g) does not add much to the story. The title is also wearing a large size pants, which actually does not capture the story very well.

We thank the reviewer for their constructive feedback, and the positive review of our work!

Specific points

1. The title is quite abstract and too large for what is reported here: the identification of the factors mediating PVM-ER contact by a forward genetic screen. Since this SPLICS system was already used before, although not in a trans-kingdom context, it does not carry sufficient novelty to justify the title.

We have changed the title to: "Identification of host ER-*Toxoplasma* contact site tethers"

2. Fig 6g (& discussion lines 376-387). This is a cool experiment, but without validation that lipid access/transfer is the mechanism, this is a bit of a distraction from the rest of this great study. Together with the discussion offering several scenarios that cannot be differentiated, my suggestion is to take it out of this manuscript to stay focused.

We thank the reviewer for this point. As suggested, we have removed this experiment from the revised manuscript.

3. Fig 6h. The growth deficiency in absence of the ER-PVM DCS. Only 1 time point at 24 hpi is shown. Only mean FI is plotted, suggesting lower parasite burden. However, this

lower burden can be due to a delayed invasion efficiency as well as slower replication, which cannot be discerned in the way the assay is set up. The flow data should contain the number of infected cells, which if the same between conditions, would hint that delayed invasion is not a problem. However, an additional time point, e.g. 12 hrs for both mean FI and % infected cells would consolidate this even more.

Good points, the IFAs you are doing now though should clarify this (with maybe the 12h time point added),

We thank the reviewer for this point. As suggested, we included the 12 hour point. Our results show that growth in VAP DKO cells is unaffected at earlier time points (indicating a defect in proliferation rather than invasion) (Fig. 4h). Furthermore, in data we did not include in the manuscript (right), we found that infection rates were similar:

Minor points

1. Fig 2a. The font in the constructs is too small to read without magnification.

We have slightly increased the font sizes, and detailed all aspects of the construct in caption fig. 2a.

2. Line 172/3. Fig 2d should be 2d here

We thank the reviewer for catching this error.

Reviewer #2 (Remarks to the Author):

The work describe in this manuscript addresses an important and timely topic by asking to study membrane contact sites (MCS) that likely contribute to host parasite interaction. The authors first established the presence of ER-parasite-vacuole MCS in the three major strains of the intracellular parasite *Toxoplasma gondii* and characterise the properties of those contacts. The authors then attempted to combine a new sensor of host-organelle-parasite-vacuole MCS with a loss-of-function CRISPR screening to discover an ER-parasite-vacuole molecular tethers. This led to the identification of molecules that are important for ER recruitment to the parasite vacuoles.

There are a few major weaknesses.

Sensor - the development of a sensor to study host-*Toxoplasma* interaction has potential to be valuable to study this and other intracellular pathogens once fully validated, however better controls are required for the split GFP sensor presented. While the TOM70KO experiment provides some evidence for a dynamic response of the sensor to cellular defects, it doesn't justify the conclusion that "Thus, our host-pathogen MCS sensor recapitulates *Toxoplasma*-mitochondria contact site biology" as its potential effect of artificial MCS could not be excluded with the current control. A more relevant control is to compare the mitochondria-vacuole MCS with and without the sensor via in the same experiments used to define and characterise MSC in the first paragraph.

We thank the reviewer for their constructive feedback. In our revised manuscript, we use electron microscopy to show that mitochondria-PV MCS are comparable between WT OMM^{GFP1-10} cells infected with *Toxo*^{mcherry} parasites and *Toxo*^{PVM/11} parasites. Furthermore, that TOM70 KO cells expressing OMM^{GFP1-10} and infected with *Toxo*^{PVM/11} parasites remain deficient for mitochondria-PV MCS (Extended Data 5).

In other systems, such as yeast, abundant membrane proteins on each studied membrane fused to a reporter have been successfully used as sensors for MCS and the same could be applied here.

We thank the reviewer for agreeing with our approach; indeed, for our OMM sensor we used the TM domain of TOM20, which is a highly expressed OMM protein (Fig. 2A). For our PV sensor, we used TgMAF1, which is also a highly expressed PVM protein (Fig. 2A).

Candidate selection - the selection of parasite rhoptry proteins as potential tethers included some biased decisions or application of arbitrary cutoffs. For example, since Gra45 had no effect on MCS the authors conclude that Gra proteins are identified due to their role in vacuole biology and insertion of the Maf protein, then continue to focus on Rop proteins. However, the same logic can apply for Rop – namely that their effect may be due to a role in vacuole biology rather than a tether role.

We thank the reviewer for raising this important point. To address the possibility that the loss of ROP1 was affecting the PVM localization of rhoptry proteins or vacuole biology more broadly, we engineered a TgROP1 (F166A) point mutant based on our AlphaFold predictions of TgROP1-VAPA interface residues; specifically, we mutated the Phe166 in the alternative 'FFAT' motif that is predicted to interact with the VAP MSP domain. We found that this mutation abolished host ER association with the *Toxoplasma* vacuole (Fig. 5a,b). Importantly,

TgROP1^{F166A} localizes to the PVM, ruling out pleiotropic effects on PVM-targeting (Extended Data 13).

Likewise, the motif used to identify VAP-interacting proteins, was arbitrarily “relaxed” to identify more candidates. Similarly, the pLDDT cutoff is arbitrary. The question is then, was the screen useful or did the authors ended up taking an educated guess. What would be the outcome if all known PMV proteins were analysed in the same way. The readers should be able to appreciate if the screen provided any added value.

We thank the reviewer for raising this point and apologize for our lack of clarity. Our pLDDT cutoff was not chosen arbitrarily but was derived from a previously established cutoff benchmark (PMID: 38225382). In this study, our co-author Katja Luck (senior author of PMID: 38225382) systematically evaluated AlphaFold-Multimer v2 predictions of domain-motif interfaces and defined optimal cutoffs to distinguish reliable from unreliable models. We applied these cutoffs to assess the presence of FFAT and relaxed FFAT motifs among candidate proteins. Thus, the screen was essential: it provided a structured, unbiased framework to triage candidates based on defined criteria. We have revised the results section to make this workflow clearer.

While the manuscript described several elegant but indirect experiment to support a potential Rop1-VAPA/B tether for host-ER-parasite-vacuole MCS (such Rop 1 dependent recruitment of VAP; MCS dependency on VAP). The work lack an experiment that demonstrate this directly.

We thank the reviewer for raising this point. In our revised manuscript, we provide several experiments that support a ROP1-VAPA/B tether for host ER-PV contact sites:

1) An interaction between ROP1 and VAPA/B: we show VAPA and VAPB, but not other ER proteins such as calnexin and HMGCR, are enriched in ROP1-IPs (Fig. 4A).

2) MCS dependency on VAP: we show that targeting VAPA to the outer mitochondrial membrane (OMM) is sufficient for MCS between host mitochondria and the PV of Type II parasites, which naturally lack such contacts because they are deficient for TgMAF1 (Fig. 5c-g). This demonstrates that VAPA, independent of other ER factors, is sufficient for MCS formation.

3) ROP1-dependent MCS: *Irop1* parasites expressing a point mutant of TgROP1 based on our AF multimer predictions of TgROP1-VAPA interface residues — we mutated the Phe166 in the alternative ‘FFAT’ motif that is predicted to interact with the VAP MSP domain — are deficient for ER association (Fig. 5a,b).

At the same time the indirect experiments lack critical controls. For example, the authors show VAP DKO results in loss of ER-vacuole association however is this specific or perhaps a general ER defect? What is the effect on other ER MCS (e.g. host ER-mitochondria MCS)? What is the effect on ER biogenesis and function?

We thank the reviewer for raising these interesting points. As aforementioned, targeting VAPA to the OMM in VAP DKO cells is sufficient for MCS between Type II and host mitochondria. Thus, VAPA, independent of other ER factors, drives ROP1-dependent MCS (Fig. 5c-g). We agree that the effect on other ER MCS and ER biogenesis is interesting and believe our identification of the factors required for host ER-*Toxoplasma* MCS allows us to address these questions in the future.

Reviewer #3 (Remarks to the Author):

Interorganellar membrane contact sites (which the authors call MCS) have radically altered our understanding of organellar function. Unsurprisingly, intracellular pathogens manipulate MCS in order to exploit their host cell's physiology. It is well-established that *Toxoplasma* has contacts with both the host mitochondria and ER. In the present study, the authors build upon some of their previous work in characterizing the *Toxoplasma* PVM-mitochondria to examine the PVM contacts with the host ER. Using an innovative biosensor and candidate-based CRISPR screen, the authors identify an intriguing parasite effector candidate in ROP1. They demonstrate convincingly that ROP1 expression is required for ER-PVM contacts, and that host VAPA/B are required on the host side.

We thank the authors for their positive review of our work and their insightful feedback.

Overall, the experiments are well described and the data are of high quality. I'm very excited by the potential here, but what strongly diminishes my excitement is concern over some sort of pleiotropic effect. Briefly, as the authors point out, GRA45 was originally identified in a very similar screen looking for proteins that altered host immune signaling, but has nothing (directly) to do with that; it appears to act far upstream, somewhere in trafficking of the proteins themselves. In the original paper describing rescue of the ROP1 knockout (pmid 8719248), the rhoptries were reported to be grossly changed in their morphology, a phenotype that was rescued by reintroduction of ROP1 expression (this was confirmed in the Butterworth et al paper from one of the co-authors labs). It is therefore possible that ROP1, like GRA45, has some sort of upstream effect that is overwhelming the screen. That ROP1 (like GRA45) has come up in multiple CRISPR screens for effectors is further cause for concern.

Unfortunately, the experiments the authors have performed do not rule out such an explanation. With a few more reasonably straightforward experiments (described below), though, I'd be quite convinced.

Other proteins appear de-enriched to at least the extent that ROP1 is in the mutant vs. wt VAPA IP (Fig 5b). Can the authors comment on any of these other proteins?

We thank the reviewer for this interesting observation. Although we have not directly investigated the other de-enriched proteins, one possibility is that additional *Toxoplasma* effectors may localize to host ER-PV MCS once they are established, thereby indirectly association with VAPA. Given that this interpretation remains speculative, we chose to remove this experiment from the revised manuscript.

In my opinion, to really nail that ROP1 is directly binding VAPA/B, the authors need to make some sort of point mutant(s) (not deletions of residues!) that do not rescue the ER-PVM MCS when expressed in $\Delta rop1$. This should be reasonably straightforward given the AlphaFold predictions.

We thank the reviewer for this helpful suggestion. To directly test this, we generated *Irop1* parasites expressing a ROP1 point mutant based on our AF multimer predictions of the TgROP1-VAPA interface (Fig. 5a,b). Specifically, that mutating Phe166 in the alternative 'FFAT' motif that is predicted to interact with the VAP MSP domain abolishes ER association (Fig.

5a,b). This result supports our conclusion that ROP1 directly binds VAPA to mediate ER-PVM MCS.

Given that both ROP1 and VAPA can easily be expressed in bacteria, the authors could further strengthen their claims of direct binding by e.g. GST-pulldown. This would also allow them an excellent platform to demonstrate/screen the relevance of any mutants.

We agree with the reviewer that GST-pulldown experiments would be an excellent platform to test direct binding. However, we were unable to successfully purify ROP1 under conditions that preserved its stability (in contrast to MAF1, which expresses readily). To nevertheless address this point, we show that Type II parasites, which lack TgMAF1 and thus do not normally tether mitochondria—form mitochondria-PV MCS when VAPA is targeted to the OMM in VAP DKO cells. This demonstrates that VAPA is sufficient to mediate MCS, independent of other ER membrane factors, and supports a direct interaction between TgROP1 and VAPA.

Another concern – given that overexpression of the mutants in VAP-A that the authors use in their study have been demonstrated to induce gross changes in ER morphology (e.g. pmid 16004875), is it possible that the effects they see in their knockout or mutants of VAP-A are also potentially secondary? Since that came from a (very reasonable) hypothesis rather than a screen, extra controls may be warranted.

We thank the reviewer for raising this important point. The overexpression of certain VAPA mutants has indeed been reported to induce changes in ER morphology. We however bypassed potential effects of ER remodeling by showing that OMM-targeted VAPA is sufficient to drive MCS formation (with Type II parasites that lack TgMAF1 and thus do not normally tether mitochondria, Fig. c-g). This result supports our conclusion that the loss of host ER-PV MCS in VAP DKO cells is due to the direct interaction between TgROP1 and VAPA, rather than a secondary effect of altered morphology.

It seems a bit odd that almost all of the experiments are performed very early after infection (~3 hours; before any replication). Does the nature of ER contacts change after the parasite has replicated 1 or more times? Can the authors add some description of their reasoning for this?

We thank the reviewer for this question. We chose to focus on ~ 3 hours post-infection because this time point allows us to capture the entire vacuole within a single EM image (with maximal resolution), enabling accurate assessment of the full PVM perimeter. Although ER-*Toxoplasma* MCS are maximal at early stages of infection (PMID: 9378762), no apparent differences in the nature of ER contacts are evident at late stages of infection.

Minor:

First off – what is the provenance of the Δ rop1 and complement strain? Its generation is not described in the methods (just Δ rop6) and it is not cited as having been received from another source (nor are references for it cited – I'm assuming the recent strains made in the Treck lab). [I ok I re-read and found the Butterworth reference here, but it was not entirely clear to me that this was the source of the relevant parasites – the source of all strains should be made clear in the methods as well]

We thank the reviewer for this comment and have made sure to cite provenance of the ROP1 KO parasites where mentioned.

- I appreciate the attention to detail in the Methods. (other than neglecting to mention where the Δ rop1 came from)

We thank the reviewer for this comment.

- Many of the color choices used to annotate images are hard to read on the black backgrounds, and in some cases (especially blues) are not color blind friendly. A white stroke would help with this (and/or choosing slightly brighter colors).

We thank the reviewer for this feedback and have tried increasing the brightness of text in images.

- In all tables featuring gene models, it would be helpful to the reader to include gene names/predictions where available (e.g. proteomics tables)

We have filled in gene descriptions where available.